



# Ship- and island-based atmospheric soundings from the 2020 EUREC4A field campaign

Claudia Christine Stephan[1], Sabrina Schnitt[2], Hauke Schulz[1], Hugo Bellenger[3], Simon P. de Szoeke[4], Claudia Acquistapace[2,*], Katharina Baier[1,*], Thibaut Dauhut[1,*], Rémi Laxenaire[5,*], Yanmichel Morfa-Avalos[1,*], Renaud Person[6,7,*], Estefanía Quiñones Meléndez[4,*], Gholamhossein Bagheri[8,+], Tobias Böck[2,+], Alton Daley[9,+], Johannes Güttler[10,+], Kevin C. Helfer[11,+], Sebastian A. Los[12,+], Almuth Neuberger[1,+], Johannes Röttenbacher[13,+], Andreas Raeke[14,+], Maximilian Ringel[1,+], Markus Ritschel[1,+], Pauline Sadoulet[15,+], Imke Schirmacher[16,+], M. Katharina Stolla[1,+], Ethan Wright[5,+], Benjamin Charpentier[17], Alexis Doerenbecher[18], Richard Wilson[19], Friedhelm Jansen[1], Stefan Kinne[1], Gilles Reverdin[20], Sabrina Speich[3], Sandrine Bony[3], and Bjorn Stevens[1]

[1]Max Planck Institute for Meteorology, Hamburg, Germany
[2]Institute for Geophysics and Meteorology, University of Cologne, Cologne, Germany
[3]LMD/IPSL, CNRS, ENS, École Polytechnique, Institut Polytechnique de Paris, PSL, Research University, Sorbonne Université, Paris, France
[4]College of Earth, Ocean, and Atmospheric Sciences, Oregon State University, Corvallis, Oregon, USA
[5]Center for Ocean-Atmospheric Prediction Studies, Florida State University, Tallahassee, Florida, USA
[6]Sorbonne Université, CNRS, IRD, MNHN, INRAE, ENS, UMS 3455, OSU Ecce Terra, Paris, France
[7]Sorbonne Université, CNRS, IRD, MNHN, UMR7159 LOCEAN-IPSL, Paris, France
[8]Laboratory for Fluid Physics, Pattern Formation and Biocomplexity, Max Planck Institute for Dynamics and Self-Organization, Göttingen, Germany
[9]Caribbean Institute for Meteorology and Hydrology, Husbands, St. James, Barbados
[10]Max Planck Institute for Dynamics and Self-Organization, 37077 Göttingen, Germany
[11]Department of Geoscience and Remote Sensing, Delft University of Technology, Delft, The Netherlands
[12]Department of Earth and Planetary Sciences, University of New Mexico, Albuquerque, New Mexico, USA
[13]Institute for Meteorology, University of Leipzig, Leipzig, Germany
[14]Deutscher Wetterdienst, Seewetteramt Hamburg, Seeschiffahrtsberatung - Bordwetterdienst, Hamburg, Germany
[15]Météo-France, Bordeaux, France
[16]University of Hamburg, Hamburg, Germany
[17]Meteomodem, Ury, France
[18]Météo-France and CNRS: CNRM-UMR 3589, 42 Av. G. Coriolis, 31057 Toulouse Cedex, France
[19]Sorbonne Universite, LATMOS/IPSL, INSU/CNRS, Paris, France
[20]LOCEAN, SU/CNRS/IRD/MNHN, Sorbonne Université, Paris, France
*These authors contributed equally to this work.
+These authors also contributed equally to this work.

**Correspondence:** Claudia C. Stephan (claudia.stephan@mpimet.mpg.de)

**Abstract.** To advance the understanding of the interplay among clouds, convection, and circulation, and its role in climate change, the EUREC4A and ATOMIC field campaigns collected measurements in the western tropical Atlantic during January and February 2020. Upper-air radiosondes were launched regularly (usually 4-hourly) from a network consisting of the Barbados Cloud Observatory (BCO) and four ships within 51–60 °W, 6–16 °N. From January 8 to February 19, a total of





812 radiosondes measured wind, temperature and relative humidity. In addition to the ascent, the descent was recorded for
82 % of the soundings. The soundings sampled changes in atmospheric pressure, winds, lifting condensation level, boundary
layer depth, and vertical distribution of moisture associated with different ocean surface conditions, synoptic variability, and
mesoscale convective organization. Raw (Level-0), quality-controlled 1-second (Level-1), and vertically gridded (Level-2) data
in NetCDF format (Stephan et al., 2020) are available to the public at AERIS (https://doi.org/10.25326/62). The methods of
10 data collection and post-processing for the radiosonde data set are described here.

## 1 Introduction

A number of scientific experiments have focused on the trade-cumulus boundary layer over the tropical Atlantic Ocean. The
Barbados Oceanographic Meteorological Experiment (BOMEX 1969; Kuettner and Holland, 1969), Atlantic Trade-Wind Ex-
15 periment (ATEX 1969; Augstein et al., 1973), Atlantic Stratocumulus Transition Experiment (ASTEX 1992; Albrecht et al.,
1995), and Rain in Shallow Cumulus Over the Ocean (RICO 2006; Rauber et al., 2007) experiment measured thermodynamic
and wind profiles of the Atlantic trade regime (reviewed by Baker, 1993). With these profiles as initial and environmental
conditions, models of the cumulus clouds explain their interaction with the environment (e.g. Arakawa and Schubert, 1974;
Albrecht et al., 1979; Krueger, 1988; Tiedtke, 1989; Albrecht, 1993; Bretherton, 1993; Xue et al., 2008; vanZanten et al.,
2011).

Arrayed networks of soundings have been used to characterize the interaction of clouds, convection, and the synoptic envi-
ronment. In many examples, they have been used to diagnose tendencies of the heat, mass, and moisture budgets for the tropical
atmosphere (e.g. Reed and Recker, 1971; Yanai et al., 1973; Nitta and Esbensen, 1974; Lin and Johnson, 1996; Mapes et al.,
2003; Johnson and Ciesielski, 2013). These experiments in the deep tropics monitored the synoptic (100–1000 km) variations
of vertical motion and moisture convergence as context for the evolution of the ensemble of convective clouds observed within
their sounding networks.

These sounding arrays measure horizontal divergence, which is used to estimate mean large-scale vertical motion. In
DYCOMS-II, Lenschow et al. (2007) used stacked flight circles to estimate subsidence on a fine scale relevant to marine
stratocumulus clouds. Studying the variations of mesoscale ($\sim$100 km) organization of the trade wind shallow cumulus clouds
likewise requires fine horizontal resolution. The Next-Generation Aircraft Remote Sensing for Validation Studies (NARVAL;
Stevens et al., 2016, 2019; Bony and Stevens, 2019) demonstrated that circles of dropsondes released from aircraft above the
shallow clouds reliably measure a snapshot of vertical motion.

The shallow trade cumulus clouds over the tropical Atlantic Ocean are a focus also of the Elucidating the Role of Clouds-
Circulation Coupling in Climate Campaign (EUREC[4]A; Bony et al., 2017) and associated campaigns, i.e, the Atlantic Tradewind





Ocean–Atmosphere Mesoscale Interaction Campaign (ATOMIC) [1]. The experimental design of EUREC[4]A involved 85 drop-sonde circles from aircraft flights combined with regular around-the-clock upper air observations from surface-launched radiosondes. The regular sampling from surface-launched radiosondes complemented the mesoscale vertical velocity measurements from dropsonde circles by continuously measuring time-height profiles of the atmosphere, synoptic variability for an extended time period, and diurnal variability. Radiosondes sampled when research aircraft were not flying, notably at night.

Between January 8 and February 19, 2020, 812 radiosondes were launched from Barbados and the northwestern tropical Atlantic Ocean east of Barbados. A focus of the campaign was on shallow cumulus clouds, their radiative effects, and their response to the large-scale environment, contributing progress toward the World Climate Research Programme's Grand Challenge on Clouds, Circulation and Climate Sensitivity (Bony et al., 2015). Other EUREC[4]A investigations focus on air-sea interactions due to ocean mesoscale eddies, cloud microphysical processes, and the effect of shallow convection on the distribution of winds. The present paper introduces the radiosonde observations and their resulting data sets.

Radiosondes were launched from Barbados and four research vessels. The island-based launches took place at the Barbados Cloud Observatory (BCO; 59.43 °W, 13.16 °N), situated at Deebles Point on the windward coast of Barbados. Surface and remote sensing observations at BCO have been in operation since April 1, 2010 (Stevens et al., 2016).

Four research vessels launched radiosondes over the northwestern tropical Atlantic east of Barbados (51–60 °W, 6–16 °N) during EUREC[4]A: two German research vessels, *Maria S. Merian* (hereafter *Merian*) and *Meteor*, a French research vessel, *L'Atalante* (hereafter *Atalante*), and a United States research vessel, *Ronald H. Brown* (hereafter *Brown*). The BCO and the research vessels all measured surface meteorology and deployed various other measurements for remote sensing of clouds and the atmospheric boundary layer.

In Section 2 we describe the measurement strategy for the coordinated EUREC[4]A radiosonde network, the data collection procedures for each platform, and the post-processing steps that were applied to create the final data set. Section 3 shows an overview and some characteristics of the data and is followed by a summary in Section 4. The *Atalante* additionally launched a different type of sonde, which is described in the appendix.

## 2   Sounding measurements

### 2.1   The EUREC[4]A sounding network

The number of launches per day as well as the dates of regular observations (Fig. 1) differ from platform to platform, reflecting availability of ships and personnel. Soundings supported specific research interests on each platform, in addition to the coordinated EUREC[4]A sounding network. We designed the radiosonde network to optimize the joint contribution of all platforms to the overarching goals of EUREC[4]A. Sounding platforms were usually spaced to optimally sample the scales of the synoptic circulation. The *Meteor* remained nearly stationary at a longitude of 57 °W and moved within a meridional corridor between 12.0–14.5 °N to support coordinated aircraft measurements in its vicinity (Fig. 2a). The *Brown* occupied a southwest-northeast

---

[1]Because the sounding network and EUREC[4]A comprised many projects, or component campaigns, we refer to the union of these simply as EUREC[4]A.





transect along the direction of the climatological surface trade winds, and approximately orthogonal to *Meteor*'s sampling line. The *Brown*'s transect between the BCO (59.43 °W, 13.16 °N) and the Northwest Tropical Atlantic Station for air-sea flux measurements buoy (NTAS) at 51.02 °W, 14.82 °N (Fig. 2b) sampled airmasses upwind of the BCO that move westward with the climatological easterly trade winds within 12.5–14.5 °N. This elongated region between BCO and NTAS is referred to as the

70 'Trade-wind Alley'. The *Merian* and *Atalante* ventured southward to a minimum latitude of ∼6.5 °N to observe oceanic and atmospheric variability associated with Brazil Ring Current Eddies as they tracked northwestward along the corridor referred to as 'Boulevard des Tourbillons'. The *Atalante* and *Merian* thus often form the southern points of the radiosonde network (Fig. 2c, d).

Aircraft operations included a circular flight pattern of 180–200 km diameter centered at ∼13.3 °N, ∼-57.7 °E. Dropsondes

were deployed along the circle to estimate the area-averaged mass divergence, as described in Bony and Stevens (2019). To sample larger scales than represented by this circle, we aimed at 4-hourly soundings from all five stations while platforms were separated by more than 200 km. The launch frequency was reduced when such a separation could not be maintained or when vessels left the key region of the network, i.e. moved south of 12 °N. These scenarios occurred from time to time in order to support other measurements. Figure 3 shows that the network sampled large scales for 30 consecutive days.

To increase the number of vertical profiles, we recorded the ascent as well as the descent of the radiosondes. Except for the *Brown*, balloons were equipped with parachutes, which nearly match fall speeds to balloon ascent speeds. Given that a typical ascent takes about 90 min, a radiosonde was sampling the air somewhere above each platform nearly continuously during regular operation. All platforms deployed Vaisala RS41-SGP radiosondes and used Vaisala MW41 ground station software to record and process the sounding data. To start a sounding, a radiosonde sensor was placed on the ground station for an

automated ground check initialization procedure, which took about 5–6 min. The frequency at which the radiosonde transmits its signal to the receiver was set manually to a designated value for each platform (listed in Table 1) to avoid radio interference.

The default launch times were 0245, 0645, 1045, 1445, 1845, and 2245 UTC, to have radiosondes reach 100 hPa at standard synoptic times (00 and 12 UTC). Departures from this schedule occurred due to a variety of reasons, including defective radiosondes, balloon bursts before the launch, collisions of ascending radiosondes with other on board instrumentation, and

90 air traffic safety. In the following section, we describe specific issues and aspects of the launch procedure particular to each platform.

### 2.1.1 Barbados Cloud Observatory (BCO)

The BCO is located at the eastern-most point of Barbados (59.43 °W, 13.16 °N) and thus directly exposed to easterly trade winds from the ocean. The BCO launched 182 sondes, of which 162 measured descents. Radiosondes were prepared inside an

95 air-conditioned office container with air temperature and relative humidity adjusted to 20 °C and 60 %, respectively. Balloons were prepared outside and placed into a launcher whose size provided rough guidance for achieving the desired filling level (Fig. 4a). Launches were coordinated with Barbados Air Traffic Control, which delayed soundings up to 15 min. Surface conditions obtained from the weather station observations at the BCO were entered into the software after automatic release detection.



### 2.1.2 R/V *Meteor*

The *Meteor* launched 203 sondes and collected data for 167 descents during the EUREC[4]A core period (January 8 to February 19). Eight additional ascents and descents, respectively, were recorded after February 20. Radiosondes were prepared inside a laboratory on the top deck of the ship with the antenna placed on the roof. Before February 9 the soundings were launched from the container of the German Weather Service (DWD) on the port side at the stern of the ship (Fig. 4b). This container had a marker to indicate the optimum fill level of the balloons.

On February 9 the DWD launcher broke and a launcher of the type shown in Fig. 4a was used, located at the stern of the ship. An awning over the balloon indicated the fill level. Ground data were obtained from on-board instruments of the DWD. In addition to sondes launched by the EUREC[4]A science crew, the DWD launched one radiosonde per day. The 31 ascending DWD sondes launched during the EUREC[4]A core period, plus an additional eight after February 20, are included in the Level-1 and Level-2 data sets, described in Section 2.3.

### 2.1.3 R/V *Ronald H. Brown* (*Brown*)

The *Brown* released 170 sondes and collected data for 159 descents. The radiosondes were initialized and ground-checked inside an air-conditioned laboratory. Near-surface measurements were recorded from the ship's meteorological sensors via the ship computer system display. The ground station antenna was located on the aft 02 deck railing above the staging bay. Initialized radiosonde sensor packages were placed for 1–5 min on the main deck to equilibrate to ambient environmental conditions and check GPS reception and telemetry. The balloons were filled by hand in the staging bay (Fig. 4d), which was mostly sheltered. Operators avoided unnecessary contact with the balloon body but restrained it by hand if the wind was strong.

On leg 1 (January 8–24) at night, less helium was used to reduce the buoyancy of the balloons in order to achieve lower ascent rates and better resolve the fine-scale vertical structure of the atmosphere. The ascent rate for day launches was $4.4\pm0.5$ m s$^{-1}$; for night launches, ascent was about 12 % slower, $3.9\pm0.6$ m s$^{-1}$. After January 24, the same target ascent rate was used for day and night, and operators obtained consistent balloon volumes by timing the filling.

Balloons were launched from a location on the deck to minimize the effect of the ship and obstructions on the sounding. The ship usually turned or slowed to improve the relative wind for the sounding. The relative wind carried the sounding away from the ship, but the ship's aerodynamic wake made the first ~5 s of the balloon's flight unpredictable. The sounding was sometimes launched up to 10 min earlier or later to accommodate other ship operations.

### 2.1.4 R/V *L'Atalante* (*Atalante*)

The *Atalante* launched 139 Vaisala sondes and measured 138 descents. A coordinated sounding phase was performed with the *Merian* to increase the temporal resolution from January 30 at 2045 UTC to February 2 at 1645 UTC around 52–54 °W and 6–8 °N. During this period launching times were shifted by 2 hours aboard the *Atalante* (0045, 0445, 0845, 1245, 1645, 2045 UTC) while the *Merian* launched at regular times. In addition to the Vaisala soundings, 47 sondes of MeteoModem type M10



were launched from the *Atalante* to measure the lower atmosphere across mesoscale sea surface temperature (SST) fronts, as detailed in the appendix.

The radiosondes were prepared aft of the bridge. This open space was right next to the top building of the ship, which may have affected measurements at low levels. Before launching, operators asked the bridge for direction change if necessary and

135 possible. The balloons were launched by hand from the rear deck of the bridge, where the launcher was situated (Fig. 4e). The Vaisala antenna was installed on the roof top. Surface measurements were obtained from local measurements on board. At the beginning of the campaign a frequency of 401.0 MHz was selected, which later on had to be switched to 401.2 MHz because of radio interference at 400.9 MHz from an unknown source. This interference caused loss of signal for two radiosondes during their ascent. When a previous sounding was not terminated at the launch time of a subsequent sounding, a frequency of 400.7

140 MHz was selected.

The *Atalante* experienced substantial instabilities of the Vaisala acquisition system at the initialization step of the system (system location unavailable) and with the reception of the GPS signal by the Vaisala antenna and radiosondes. These problems required multiple restarts of the software and the acquisition system (between 1 and 8 times), creating delays between 10 min and 1 h. However, they did not affect the quality of the soundings. The operators checked the cables and replaced the GPS

antenna of the Vaisala system with an antenna that had a larger DC voltage range (15 V instead of 4 V). Nevertheless, the problems persisted during the cruise with the need to restart the system several times before each launch.

### 2.1.5 R/V *Maria S. Merian* (*Merian*)

The *Merian* launched 118 sondes and recorded 38 descents. Fewer sondes were launched on the *Merian* than other platforms (Fig. 1) due to difficulties and priority of *Atalante* sondes when the ships were close to one another. The radio signal was often

lost using the first antenna location, which the team suspected was due to blocking by the chimney. A new location improved the reception of the signal.

The *Merian* was equipped with a launch container (Fig. 4c). The helium fill level was decided by inflating the balloon until it reached the upper edge of the launch container. During the day, temperatures in the container rose considerably higher then ambient, but the container was well ventilated as the launch was prepared, such that the instruments experienced typical

temperatures of 28–31 °C during synchronization, with only few exceptions. Nonetheless, the residual warming could be a source of bias relative to the surface meteorology observations and persist for tens of meters after the launch. Near-surface data were taken from ship measurements.

### 2.2 Real-time sounding data distribution

Sounding observations distributed in real-time over the Global Telecommunication System (GTS) improve atmospheric anal-

160 yses for initializing and verifying weather forecasts, and improve subsequent reanalyses. Therefore, we aimed to disseminate as much of the full 1-second resolution radiosonde data from the EUREC$^4$A campaign as possible over the GTS. Radiosonde data (ascent and descent) from the *Atalante* (114 reports during the campaign) and the BCO (60 reports in February) were sent to the GTS through a Météo-France entry point. This allowed their assimilation in numerical weather prediction (NWP)





systems. Most of the *Brown* data were sent to the US National Center for Environmental Prediction (NCEP). From here they were ingested into US Weather Service and Navy NWP systems, yet not European ones. None of the data from the *Merian* and *Meteor* could be transmitted to the GTS by satellite internet. However, during EUREC⁴A, 29 daily ascent soundings from the *Meteor* were sent to the GTS via the EUMETNET Automated Shipboard Aerological Program (E-ASAP), at around 1630 UTC.

World Meteorological Organization Binary Universal Form for the Representation of meteorological data (BUFR) were submitted to the GTS and exchanged among the platforms during the EUREC⁴A campaign. BUFR supports ascending soundings (BUFR 309057), descending soundings (BUFR 309056, since BUFR Table version 31.0.0), and dropsondes released from aircraft (BUFR 309053). The Vaisala MW41 sounding software writes quality-controlled BUFR files. The sounding instruments measure relative humidity, but the BUFR files only contain the derived dew point temperature. We obtain the relative humidity from the dew point by inverting the dew point formula exactly.

## 2.3 Quality control and data formats

The Vaisala RS41 temperature and humidity measurements are highly robust and accurate, even in cloudy environments. The humidity sensor is actively heated to prevent water condensation and frost formation on the sensor surface. The Vaisala MW41 software writes proprietary .mwx binary files which are ZIP-archives that contain both the raw as well as the processed measurements. These data make up our Level-0 data set. We also provide Level-1 and Level-2 data, which we describe in the following. Our assignment of levels for the data sets adheres to the standards laid out in Ciesielski et al. (2012).

### 2.3.1 Level-1 data

Level-1 data in NetCDF format are quality controlled and averaged to 1-second resolution from the Level-0 data. Because the pressure, temperature and humidity are measured with a different sensor (PTU) than wind and position, the data are synchronized to the PTU time. This synchronization is done by the Vaisala MW41 software and the results are included in the Level-0 archive files. The Level-1 data were processed from these results.

The Vaisala MW41 sounding system applies a radiation correction to daytime temperature measurements by subtracting increments that vary as a function of pressure and solar zenith angle. The uncertainty of the radiation correction is typically less than 0.2 °C in the troposphere; uncertainty gradually increases in the stratosphere.

The Vaisala system applies algorithms to adjust for time lags of the RS41 sensors. At 10 hPa the response time of the temperature sensor is 2.5 s for an ascent speed of 6 m s⁻¹. At 18 km (75 hPa) with a temperature lapse rate of 0.01 °C m⁻¹ and an ascent rate varying from 3 to 9 m s⁻¹, the remaining uncertainty in the temperature reading due to time lag is 0.02 °C. At lower altitudes the uncertainty is even smaller. A time-lag correction is also applied to measurements of humidity. The response time of the humidity sensor is dependent on the ambient temperature. For example, at an ascent rate of 6 m s⁻¹ and at 1000 hPa it is <0.3 s for +20 °C and <10 s for -40 °C. The remaining combined uncertainty during the sounding is 4 % relative humidity.





After time-lag adjustments, the Vaisala MW41 quality control algorithm detects outliers and smooths the data to reduce noise. Our software (Schulz, 2020a) reads the processed Vaisala mwx, and MeteoModem BUFR files, and converts them to self-describing NetCDF files. We also add the ascent or descent rate, calculated from the geopotential height and time information between consecutive measurements, to the NetCDF files. The resolution of the measurements is 1 s. The resulting
NetCDF files are the Level-1 data set distributed here.

### 2.3.2 Level-2 data

To facilitate scientific analyses, Level-2 data are provided on a common altitude grid with bin sizes of $10\,\mathrm{m}$, by averaging the Level-1 data. Mean temperature, wind components, position, and logarithm of pressure are directly averaged within bins. Relative humidity is calculated from the mean of the Level-1 water vapor mixing ratio, calculated from the water vapor pressure
formula of Hardy (1998), which is also used by the ASPEN software for EUREC[4]A dropsonde measurements.

In case of missing data within a sounding, we linearly interpolate gaps of up to $50\,\mathrm{m}$. Gaps larger than $50\,\mathrm{m}$, as well as data below $40\,\mathrm{m}$ in our Level-2 data set originating from the ship soundings, are filled with missing values. Yoneyama et al. (2002) found ship influences on radiosonde measurements to extend no further than $40\,\mathrm{m}$ above the deck. For descending soundings the raw data near the surface are missing as the signal is lost due to Earth's curvature at $300\,\mathrm{m}$ to $800\,\mathrm{m}$ above mean sea level.
The median of the lowest descent measurement is at $340\,\mathrm{m}$.

## 3 Data characteristics

### 3.1 Ascending versus descending soundings

We begin with an examination of instrument ascent and descent speeds for the different platforms (Fig. 5). The median ascent speed in the mid-to-upper troposphere is between 4.5 and $5\,\mathrm{m\,s^{-1}}$ for radiosondes launched from the BCO, *Atalante* and *Merian*
(Fig. 5a, g, i). Radiosondes launched from the *Meteor* and the *Brown* ascended at slightly smaller rates of about $4\,\mathrm{m\,s^{-1}}$ (Fig. 5c, e). For all platforms and at all altitudes the 10th and 90th percentiles are roughly symmetric about the median ascent rate and fall mostly within $\pm 1\,\mathrm{m\,s^{-1}}$ of the median. Radiosondes from the *Atalante* and *Merian* appear to have experienced stronger updrafts in the upper troposphere. This is consistent with sampling the more convectively-active conditions in the south, where there is a warmer ocean surface, more precipitable water, deeper convection and a greater chance of land influences. Above
$20\,\mathrm{km}$, the median ascent rate and the spread in ascent rates increase for all platforms.

Descent speeds exhibit a much stronger functional dependence on altitude (Fig. 5b, d, f, h, j). For platforms that employed parachutes (BCO, *Meteor*, *Atalante* and *Merian*), descent rates decrease towards the ground to a minimum of about $5\,\mathrm{m\,s^{-1}}$ in the lowest kilometers. Instruments without a parachute from the *Brown* have descent rates of sightly less than $15\,\mathrm{m\,s^{-1}}$ in the lowest few kilometers. The positive skewness of the distributions associated with stations that used parachutes is due to
225 descending radiosondes with broken or detached parachutes, or with unexpected behavior of the torn balloon remains. With





the exponential decrease of air density with altitude, descent rates increase non-linearly and rapidly with altitude, exceeding 20 m s$^{-1}$ between 20–25 km when parachutes were used and exceeding 40 m s$^{-1}$ in case of the *Brown*.

Fig. 6 compares the measurements of horizontal wind speed, air temperature and relative humidity between ascending and descending soundings. We do not expect perfect agreement between ascending and descending soundings, for several reasons. First, the instruments drift substantial horizontal distances and hence systematically sample a downwind location (as illustrated in Fig. 11f for the BCO). Second, there are variable time lags of the order of a couple of hours between ascending and descending measurements. We also note that the number of descent profiles available for computing statistics is in some cases substantially smaller than the number of ascent profiles (Fig. 1). The numbers of available measurements are again listed on the left hand side of Fig. 6. All quantities shown in Fig. 6 are computed from matched ascent-descent pairs of the same instrument.

Measurements of horizontal wind speeds do not show statistically significant differences between ascent and descent (the mean lies within the 95 % confidence intervals), with the exception of the *Brown*. Here, wind speeds at around 20 km altitude are stronger for the ascent. This systematic difference could be related to excessively rapid descent rates. Similar results are found for measurements of air temperature (Fig. 6b, d, f, h, j). In case of the *Brown*, stratospheric temperature observations during descent are warmer by more than 1 °C, suggesting a bias due to high descent rates. The same bias exists for the other platforms, but the effect is smaller and not statistically significant at the 95 % confidence level. Differences in relative humidity are not statistically significant inside the troposphere.

### 3.2 Synoptic conditions

We first present the synoptic situation for the region defined by the *Meteor* and the BCO soundings. Our initial analysis focuses on the soundings for these two platforms because they define a more or less fixed geographic area – radiosondes launched from the *Meteor* were almost all launched between 12.5 °N and 14.5 °N along 57.15 °W – bounding the subdomain that was most intensively sampled. A comparison between twelve BCO soundings with coincident and nearly co-located ship-based soundings (ships were positioned just offshore of the BCO) showed no evidence (Fig. A4) of a systematic influence of the island on the BCO soundings. Hence, the BCO soundings appear representative of the western most boundary of the marine measurement area. Focusing on a fixed region during the period of most intensive airborne operations, between January 20 and February 17, also provides a reference for quantifying differences in soundings taken outside of this region, or time period, as is discussed at the end of this subsection.

Synoptic differences among variables believed to be important for patterns of low-level cloudiness suggest that: (i) the *Meteor* and the BCO sample the same synoptic environment; and (ii) that changes in the environment can usefully be described by week-to-week variability over the four weeks starting on Monday, January 20. Fig. 7 shows that the lower-tropospheric stability, the near surface winds, the lifting condensation level (LCL) of near-surface air, and the hydrolapse associated with the depth of the trade-wind layer, as measured from the *Meteor* or BCO soundings, track each other well. Fig. 8 further illustrates that the LCL tracks well the lowest cloud bases as measured by the *Meteor* ceilometer. Week-to-week variations as deduced from the soundings of either platform show the first and last week to be characterized by a deeper moist layer, and lessened lower tropospheric stability, the latter primarily explained by changes in the potential temperature at 700 hPa.



The two week period starting on January 27 has a much shallower trade-wind layer and stronger stability. Near surface winds vary somewhat out of phase with the moisture variability, with winds stronger in the second half of the four week period, and weaker in the first half. The LCL shows very little synoptic variability.

Cloud observations are also included in Fig. 7. Reports of mid-level ($C_M$) and high-level ($C_H$) clouds are derived from 3 hourly SYNOP observations reported by the Barbados Meteorological Service at Grantley Adams International Airport. If a
265 reported mid or high-level cloud type was persistent through the day (more than three reports) it is included via its WMO cloud symbol[2] in Fig. 7. Notable are mid-level clouds that coincide with the deepening of the marine layer, particularly during the period at the end where a layer of altocumulus ($C_M = 4$) persisted for several days (Fig. 8). Observations of low clouds ($C_L$) indicated that $C_L=8$ and $C_L=2$ where the dominant low-level clouds; both evident on almost every day with little evidence of synoptic variability. This is also evident from the *Meteor* ceilometer measurements (Fig. 8). For this reason, in Fig. 7 we
instead identify days when particular patterns of mesoscale variability were in evidence. We adopted the four patterns, Sugar, Gravel, Flowers, Fish following Stevens et al. (2020) and whether or not one particular pattern was identified was taken from a cloud classification activity organized by one of the authors (H. Schulz). These patterns suggest that the moist initial period has the satellite presentation of Fish, and that the period of increased lower-tropospheric stability and strengthening winds on February 2 was associated with the pattern Flowers, consistent with the analysis of Bony et al. (2020).

To give a better impression of the synoptic variability, the period identified with the Fish pattern, between January 22–24, is investigated further. The visible satellite imagery from MODIS on Aqua (Fig. 9a) illustrates the large-scale characteristics of the observed Fish cloud pattern, covering the BCO and the northern latitudes of the observations region. The pattern resembles a spine in a surrounding cloud-free area and was accompanied by unseasonally large amounts of surface precipitation. Fig. 9b illustrates the moistening of the atmosphere and the deepening of the boundary layer, as measured at the BCO, over the course
of this event. Between January 20–26, the increase of integrated moisture up to $55\,\mathrm{kg\,m^{-2}}$ coincides well with the deepening moist layer, thus also with changes in cloud top height and trade wind inversion height. Before and after the event, the inversion layer height was around $2\,\mathrm{km}$ (Fig. 7), and the boundary layer was characterized by a mixture of Gravel and Sugar, albeit the latter not on a scale that lent itself to identification from the satellite imagery. During the peak of the event on January 22 and 23, the moisture layer deepened up to 5 km. While the Fish cloud pattern passed over BCO, the pressure in the boundary
layer decreased by up to 4 hPa (see Fig. 11e) and the temperature in the upper middle troposphere ($6\,\mathrm{km}$ to $8\,\mathrm{km}$) showed a slight positive anomaly (see Fig. 11a). The rain intensity, measured at BCO with a Vaisala WXT-520 ground station, peaked at $15\,\mathrm{mm\,h^{-1}}$, and precipitation events were persistent, in contrast to the short rain showers more typical of the dry season (Stevens et al., 2016). Bony et al. (2020) found that the Fish cloud pattern often occurs under weaker surface trade wind speeds below 8 m s$^{-1}$; the sounding data confirm this, as the measured wind speeds lie well below this threshold in the lower boundary
layer, e.g. Fig. 7.

Given that the vertical structure of the humidity field appears to be a strong indicator of synoptic variability, time-height humidity plots for all of the platforms are used to explore the coherence of synoptic conditions sampled by individual platforms. This analysis (Fig. 10) shows that soundings from the *Brown*, which moved around more, but stayed mostly north of 12.5 °N

---

[2]These symbols are taken from the 2017 edition (Table 14) of the WMO Cloud Atlas (www.wmocloudatlas.org).





and east of the *Meteor*, sampled a similar synoptic environment. The *Merian* and *Atalante* however were further south and their soundings show a humidity structure and evolution that is less coherent than seen by the ships in the Trade-wind Alley. Based on this finding and because performing the same analysis for any one station does not change the big picture, we composite the soundings from all of the platforms north of 12.5 °N. Figure 11 shows the temporal evolution of atmospheric conditions for the full period of data coverage averaged north of 12.5 °N, i.e., over the Trade-wind Alley. Before January 22 the mid-troposphere is relatively cool and zonal winds in the upper troposphere are strong. From January 22 onward the observational domain experienced warmer temperatures, weaker upper-tropospheric westerlies, as well as weaker easterlies near the surface. Positive pressure anomalies first appear in the upper troposphere and reach the surface at the end of January when a ridge starts to dominate the area. Surface and upper-tropospheric winds strengthen again after February 6 when the positive pressure anomaly fades. A strong moistening of the mid and upper levels is seen around February 13, which coincides with a directional change of the meridional winds at these levels, favoring the aforementioned extensive and persistent altocumulus cloud layer (Fig. 7).

Most differences between the structure of the atmosphere within the Trade-wind Alley (North of 12.5 °N) and the 'Boulevard des Tourbillons' (southern corridor) are confined to the structure of the lower-tropospheric humidity. South of 12.5 °N, the atmosphere was on average much more humid in the lower and middle troposphere, as shown in Fig. 12. This humidity anomaly is not persistent, as dry conditions, similar to those observed north of 12.5 °N, were also present; it can rather be associated with more frequent periods of a deep moist layer and deeper convection, for example as observed during the period around January 29 (see Fig. 10). Additional, albeit less substantial differences (not shown), are that middle-upper troposphere relative humidities (between 7 km to 10 km) are actually somewhat drier in the South. There is very little evidence of systematic differences in the temperature structure between the northern and southern soundings, except for a hint of enhanced stability in the upper troposphere (11 km to 15 km) in the North. Over the 'Boulevard des Tourbillons', the depth of the near surface easterly layer is 1 km to 2 km shallower and between 5 km to 15 km, the westerlies have a stronger northerly component.

## 4 Summary

The EUREC⁴A field campaign during January–February 2020 included among its wide range of observational platforms an extensive radiosonde network, consisting of the Barbados Cloud Observatory and four research vessels. 182 radiosondes of type RS41-SGP were successfully launched in a regular manner between January 16 and February 17 from the BCO, 203 between January 18 and February 19 from the *Meteor*, 170 between January 8 and February 12 from the *Brown*, 139 between January 21 and February 16 from the *Atalante*, and 118 between January 20 and February 19 from the *Merian*. In addition, 47 MeteoModem radiosondes of type M10 were launched from the *Atalante* during intensive observational periods to sample variability associated with sea surface temperature fronts. These are described in the appendix.

We made data at three stages publicly available. Level-0 data contain the raw .mwx binary files, which can be read and processed with the MW41 software. Level-1 data were subject to Vaisala's standard quality control algorithm, which detects outliers in the profiles, performs a smoothing to reduce noise, and applies time-lag and radiation corrections. The Level-1 file



format is NetCDF with a temporal resolution of 1 s. To facilitate scientific analyses, Level-2 data are vertically gridded by averaging Level-1 data in 10-m bins. All soundings, ascending and descending, from each platform were collected into one NetCDF file for the Level-2 data.

The *Meteor* and the *Brown* followed nearly-orthogonal sampling lines, mostly in the latitude band 12.5–14.5 °N, whereas the *Atalante* and *Merian* sampled conditions further to the south. It was a central goal of EUREC[4]A to better understand the formation and feedbacks of different patterns of shallow cumulus clouds. We were fortunate that Nature provided us with a wide variety of cloud conditions, which are reflected in the radiosonde data. The six weeks of sounding data at high temporal resolution should render the radiosonde data described herein useful for a large variety of scientific analyses.

## 335    5    Code and data availability

Raw Level-0 data consist of single files per sounding in .mwx. format, which combine ascent and descent from each instrument. Quality-controlled Level-1 data consist of single files per sounding in NetCDF format, with separate files for ascent and descent. Level-2 data are stored in a single file per station and include data on a 10-m vertical resolution grid, including all available ascents and descents. Ascent and descent can be distinguished by a flag that indicates the direction. All data (Stephan et al.,
2020) are archived and freely available for public access at AERIS (https://doi.org/10.25326/62). Our software, which we used to convert to NetCDF format is also publicly available (Schulz, 2020a; https://doi.org/10.5281/zenodo.3907257) and uses the ecCodes library (https://github.com/ecmwf/eccodes).

### Appendix A:    Extra soundings on board the *Atalante*

In addition to the regular Vaisala soundings, further soundings were performed from the *Atalante* primarily to sample the
lower atmosphere across sea surface temperature (SST) fronts associated with oceanic mesoscale dynamics. An independent radiosonde receiver was used to not interfere with the regular soundings depicted in this article. MeteoModem M10 radiosondes were chosen for availability and cost. In order to decide the period of intensive sampling using these sondes, we first identified on a daily basis the ocean mesoscale eddies and currents by applying the TOEddies detection algorithm (Laxenaire et al., 2018) to the Ssalto/Duacs Near Real Time (NRT) altimeter products (Absolute Dynamic Topography – ADT – and the associated
surface geostrophic velocities; Ablain et al., 2017, Taburet et al., 2019).

These data were successively analyzed together with the Near Real Time (NRT) SST produced by Collecte Localisation Satellites (CLS), the ship's ThermoSalinoGraph (TSG) 5 m-depth temperature measurements, and ARPEGE and ECMWF forecasts in order to decide in real time the launching strategy. The NRT CLS SST is produced as a 1-day average, high-resolution product, which is a simple data average of the satellite measurements taken over the previous day, and has a resolution
of 0.02 ° in latitude and longitude. This product may have local gaps due to the presence of clouds or missing data.





Precisely setting the sounding periods was difficult because the satellite observations were only available for the previous day with additional uncertainties in the location of SST fronts due to cloud screening. Furthermore, this strategy was defined in coordination with the *Merian* to take into account the oceanographic observation goals common to both ships.

The first targeted and intensive radiosonde observation leg took place on January 26. 11 MeteoModem sondes were launched
while crossing a SST front associated with a relatively cold filament (-0.5 to -1 °C SST anomaly) steered from the Guyana coast by a mesoscale anticyclonic eddy (Fig. A1a). During this leg, the ship crossed a front of about 0.5 °C extending over 30 km with near surface wind of 6–7 m s$^{-1}$ magnitude and 60°–70° direction. During this leg the ship was heading eastward, almost into the wind. Figure A1a shows the February 25 SST map, chosen as clouds prevented retrievals on the following day. According to the satellite product, one would have expected to meet the front further east. Fortunately, a first diagonal transect
during the night provided us with the actual front location.

The second targeted and intensive radiosonde observation leg took place on February 2–3. This leg lasted for about 24 hours during which 28 MeteoModem radiosondes were launched while the ship was zigzagging in order to sample several times the northeastern edge of a cool SST anomaly of nearly -1 °C associated with coastal upwelling off the Suriname and French Guyana coast (Fig. A1b). During this leg, the ship was moving westward and sampled SST variations of 0.3–1 °C extending
over 50–60 km. At this time the near surface wind was variable in direction, 40°–80°, and relatively strong (8–11 m s$^{-1}$).

The remaining MeteoModem radiosondes were launched on few diverse occasions: two were launched in the center of the warm core of a second eddy on January 27. Another radiosonde was launched under a convective system on February 10. The last four launches took place in cloud streets on February 17.

We used M10 GPS radiosondes with an SR10 station and EOSCAN (1.4.200306) software. Only ascent data are available
for these soundings as no parachute was used and most of the launches were stopped manually at about 10 km height to increase the sampling frequency of the lower atmosphere in regions characterized by SST fronts. Launch frequencies reached up to one sounding every 40 min during the intensive launch periods. Therefore, several radiosondes were emitting at the same time, so frequencies had to be changed within the 400.4–403.4 MHz band to avoid interference. M10 radiosondes measure relative humidity and temperature, from which dew point temperature is deduced. The altitude and horizontal displacements
of the radiosondes are measured by GPS and are used to diagnose the horizontal wind components. The pressure is deduced from the altitude and the surface station pressure measurement, using the hydrostatic approximation. Our published data formats, NetCDF and ASCII formatted files (.cor files), both contain data reported every second. Note that in Level-1 data, constructed from BUFR reports that do not contain relative humidity, the latter is deduced from the dew point temperature using the Magnus-Tetens formula and might therefore slightly differ from the value in the raw .cor files that provide the direct
measurement of the radiosondes.

Figure A2 illustrates the outcome of these targeted and intensive radiosonde observations with results from the February 2–3 intensive observation period (Fig. A1b). Profile color (Fig. A2a–c) denotes the SST measured by the ship at the time of the launch (Fig. A2d). Blue (red) profiles are thus on the cold (warm) side of the SST front. These profiles are from raw data (level-0) and no attempt was made to validate, correct or remove doubtful data such as the surprisingly cold layer between
800–900 m altitude that can be seen in one of the blue potential temperature profiles (Fig. A2a). No attempt has either been





made to disentangle diurnal or synoptic scale variability from the imprint of the SST front on the lower atmosphere. However, one can note that the warm side of the SST front was sampled mostly during nighttime (local noon at 1530 UTC, nighttime from 22–10 UTC). There is a clear tendency for warmer boundary layers over the warm side of the front than over the cold side (Fig. A2a). On the other hand, the height of the mixed layer, that can be defined as near homogeneous potential temperature layers close to the surface, tends to be deeper over the cold side than over the warm side. This contrasts with results obtained over stronger SST fronts from observation (Ablain et al., 2014) and modeling studies (e.g., Kilpatrick et al., 2013; Redelsperger et al., 2019) and suggests that the lower atmosphere does not solely respond to the SST gradient. Over the cold side, wind speed tends to decrease with altitude (Fig. A2b). Over the warm side, and despite a larger variability from a profile to another, the wind speed tends to be more homogeneous in the vertical than on the cold side. Because the mixed layer depth is shallower over the warm side, it is however difficult to interpret this as the result of a stronger vertical turbulent mixing. Overall, near surface wind speed tends to be slightly weaker on the warm side than on the cold side. There is also a noticeable change in wind direction throughout the boundary layer from E-NE over the warm side to NE over the cold side (Fig. A2c).

Finally, we provide a first assessment of the quality of MeteoModem M10 measurements based on the *Atalante* soundings, as also Vaisala soundings were launched during the intensive MeteoModem periods. We compare MeteoModem and Vaisala wind, temperature and relative humidity profiles for 8 pairs of soundings that were launched within 25 min (Fig. A3). Choosing such a small time period certainly limits the number of difference profiles that can be computed, but it ensures that the two radiosondes sampled comparable situations. Mean difference profiles and corresponding standard deviations are computed on 100 m bins. Neither horizontal wind components (Fig. A3a, b) nor temperature (Fig. A3c) show any clear bias, although the differences between MeteoModem and Vaisala can be a few m s$^{-1}$ for the wind components (standard deviation of about 0.5–1 m s$^{-1}$) and about 1 °C for temperature (standard deviation of about 0.1–0.2 °C). On the other hand, despite a large noise below 4 km height, relative humidity shows a rather homogeneous moist bias of about 5 % (1–5 % standard deviation) in MeteoModem measurements compared with Vaisala (Fig. A3d). No correction was applied, neither to the temperature nor to the relative humidity measurements. In particular, corrections for the relative humidity seem necessary but are still a matter of research. An example of such corrections, developed for soundings in the continental mid-latitude can be found in Dupont et al. (2020).

*Author contributions.* Sandrine Bony and Bjorn Stevens designed the sounding strategy, which was then refined and realized in cooperation with Claudia C. Stephan and Simon P. de Szoeke. Sabrina Speich (cruise lead *Atalante*) designed, with Gilles Reverdin, and managed the measurements on board the *Atalante*. Stefan Kinne (cruise lead *Meteor*) and Friedhelm Jansen (responsible for BCO operations) managed the logistics of purchasing and transporting the radiosonde equipment. Benjamin Charpentier and Richard Wilson processed the MeteoModem data. Alexis Doerenbecher investigated the data flow through the GTS. The majority of radiosonde launches were performed by Gholamhossein Bagheri, Tobias Böck, Alton Daley, Johannes Güttler, Kevin C. Helfer, Sebastian A. Los, Almuth Neuberger, Andreas Raeke, Maximilian Ringel, Markus Ritschel, Johannes Röttenbacher, Pauline Sadoulet, Imke Schirmacher, M. Katharina Stolla and Ethan Wright. Claudia Acquistapace, Katharina Baier, Thibaut Dauhut, Rémi Laxenaire, Yanmichel Morfa-Avalos, Renaud Person and Estefanía Quiñones Meléndez played a leading role in the experimental execution. Hugo Bellenger and Simon P. de Szoeke contributed to the design of the radiosonde





network. Hauke Schulz and Sabrina Schnitt processed, quality-checked and analyzed the data. Claudia C. Stephan prepared the manuscript with contributions from all co-authors.

*Competing interests.* There are no competing interests.

*Disclaimer.* TEXT

*Acknowledgements.* We acknowledge Olivier Garrouste and Axel Roy from Météo-France for lending the MeteoModem station and launcher
and their technical assistance, and Thierry Jimonet and Météo-France Guadeloupe for their help. We would like to thank Rudolf Krockauer and Tanja Kleinert (DWD), Bruce Ingleby (ECMWF) and Jeff Ator (NOAA) for helping to track the radiosonde data sent to the GTS. We would like to thank Sophie Bouffies-Cloché (IPSL) and Florent Beucher (MF) for making ECMWF and ARPEGE forecasts available in real time. We thank Philippe Robbe, captain of the French *Atalante*, and the crew that helped us in completing successfully all the planned observations. We are grateful to Doris Jördens for the technical support with the Vaisala equipment, and to Ingo Lange, Harald Budweg, and
Rudolf Krockauer for their help with training our operators. We would like to thank the numerous volunteers performing launches at BCO, and Andreas Kopp, Joachim Ribbe, Klaus Reus, Diego Lange, Alex Kellmann and Nicolas Geyskens for launches on the research vessels. We would like to express our special gratitude to Angela Gruber for resolving any problems before they became problems. The ceilometer data from the *Meteor* (Jansen, 2020b) and surface meterology data from the BCO (Jansen, 2020a) are publicly available. We acknowledge the use of imagery from the NASA Worldview application (https://worldview.earthdata.nasa.gov), part of the NASA Earth Observing System
Data and Information System (EOSDIS). The Ssalto/Duacs altimeter products were produced and distributed by the Copernicus Marine and Environment Monitoring Service (CMEMS) (http://www.marine.copernicus.eu). The Collecte Localisation Satellites (CLS) SST and Chlorophylle-a catsat products (https://www.catsat.com/ocean-data/) were made available to us during the cruise in the framework of the Données et Services pour l'Océan (ODATIS) French national data infrastructure. The *Atalante* cruise was funded under the EUREC[4]A-OA project by the following French institutions: the national INSU-CNRS program LEFE; the French Research Fleet; Ifremer; CNES; the
Department of Geosciences of ENS through the Chaire Chanel program; MétéoFrance. Vaisala radiosondes were funded by the Max Planck Society, and the US National Oceanic and Atmospheric Administration Ocean and Atmospheric Research grant number NA19OAR4310375. MeteoModem radiosondes were funded thanks to Caroline Muller by the European Research Council (ERC) under the European Union's Horizon 2020 research and innovation programme (Project CLUSTER, grant agreement No. 805041). Claudia C. Stephan was supported by the Minerva Fast Track Program of the Max Planck Society.



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



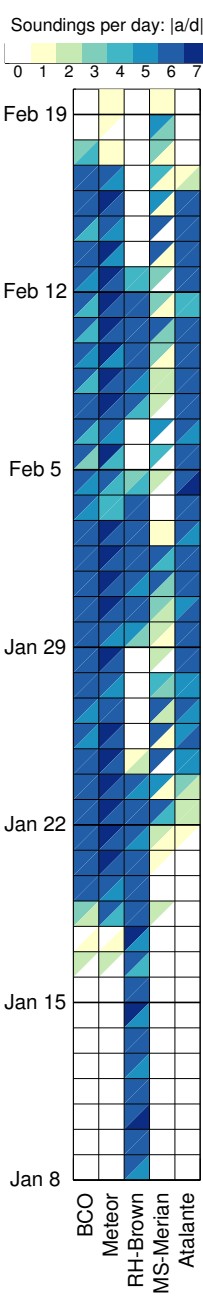

**Figure 1.** Daily number of ascending (upper left triangles) and descending (lower right triangles), respectively, soundings associated with each platform.

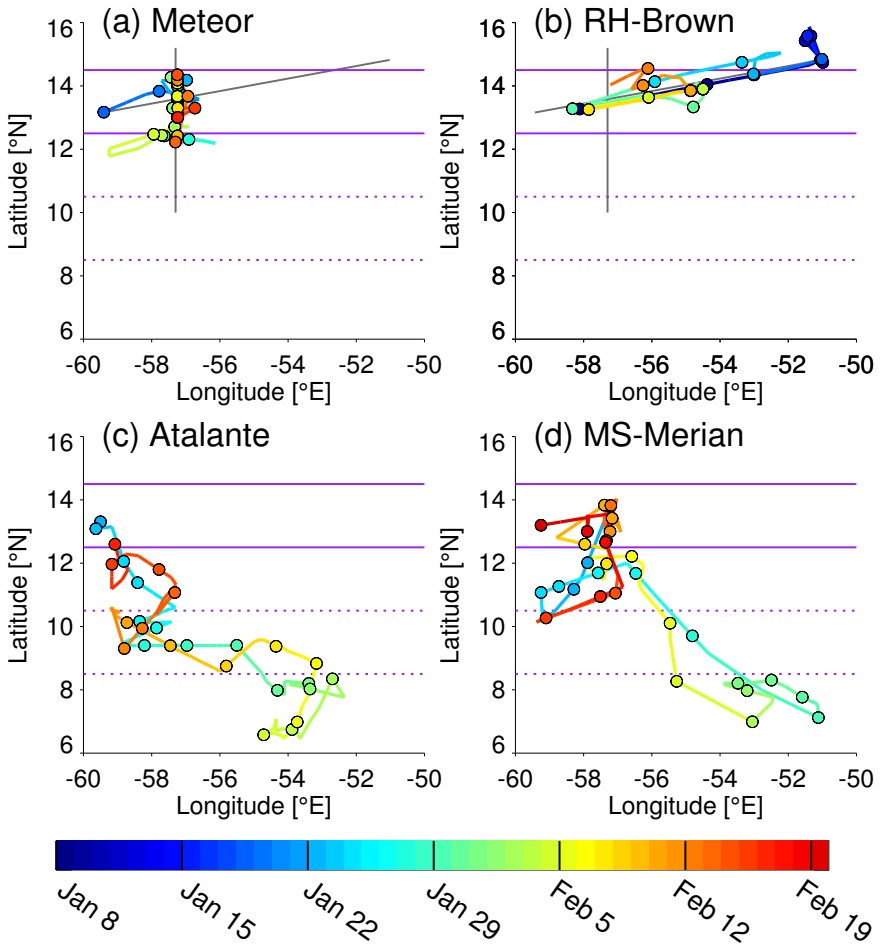

**Figure 2.** Routes and launch coordinates of radiosondes for the four research vessels colored by date. Circles mark the locations of the first radiosonde launch on each day. The gray lines in (a) and (b) mark the nearly orthogonal lines that were sampled by the *Meteor* (North–South) and the *Brown* (West–East). Purple lines mark the northern (12.5–14.5 °N; solid) and southern (8.5–10.5 °N; dashed) latitude bands that we later use to define a North (Trade-wind Alley) and South (Boulevard des Tourbillons) domain.

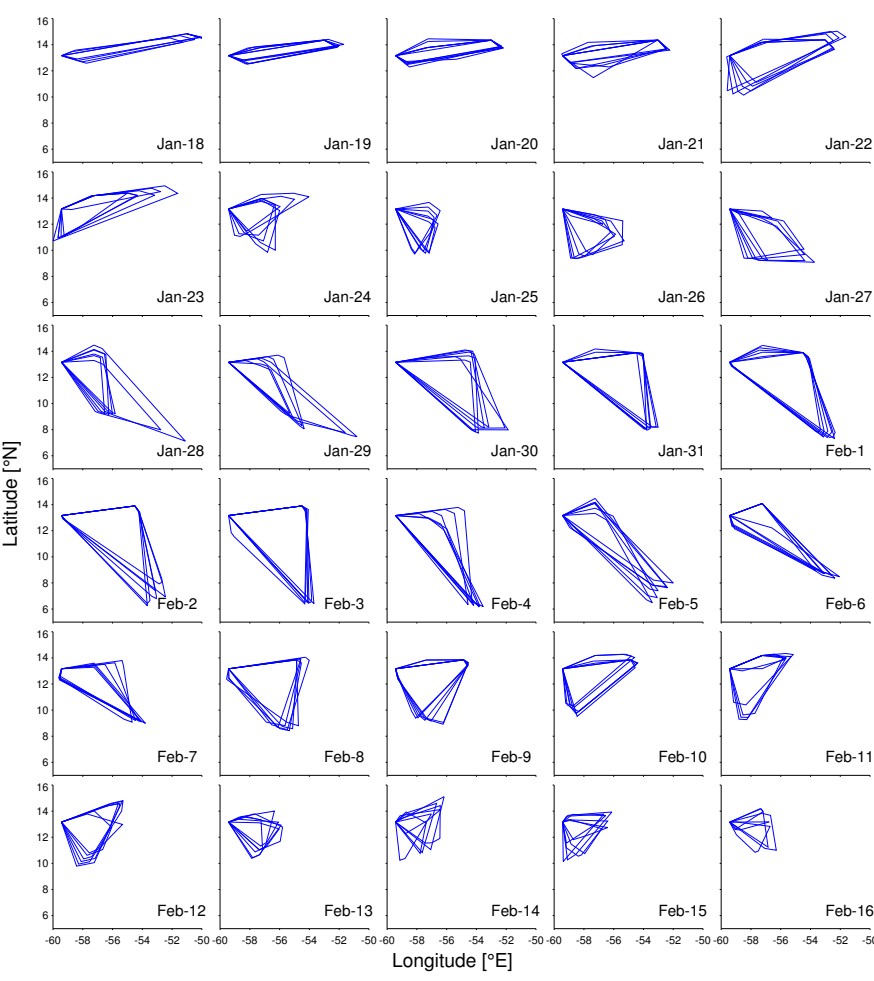

**Figure 3.** For each day between Jan-18 and Feb-16, 4-hourly polygons mark the outer bounds of the radiosonde network. Polygon vertices correspond to starting locations of either ascending or descending soundings that occurred within ±2 hours of a fixed time.

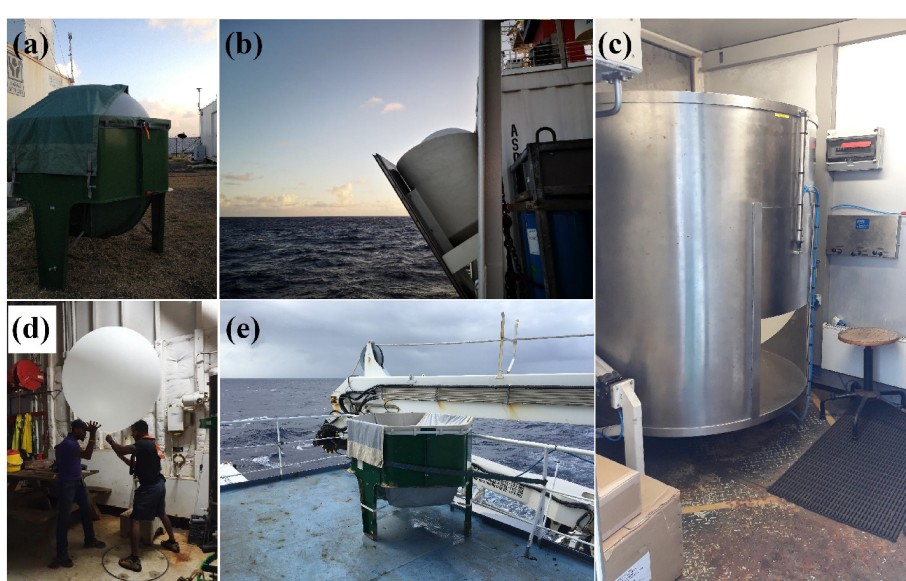

**Figure 4.** Photographs of the (a) launcher with balloon at the BCO, (b) DWD launcher with balloon on board the *Meteor*, (c) launch container with balloon on board the *Merian*, (d) manual balloon filling procedure on board the *Brown*, (e) empty launcher on board the *Atalante*.





**Figure 5.** Instrument (left) ascent and (right) descent speeds as a function of height. The sum of occurrence frequencies in each altitude bin is 100 %. The pink line shows the median profiles and the pink-green lines show the 10th and the 90th percentiles, respectively. Altitude bins are 500 m deep and speed bins are 1 m s$^{-1}$ wide. The numbers of radiosondes that crossed the corresponding height-levels (2.5, 7.5, 12.5, 17.5 and 22.5 km, respectively) are shown in each panel.

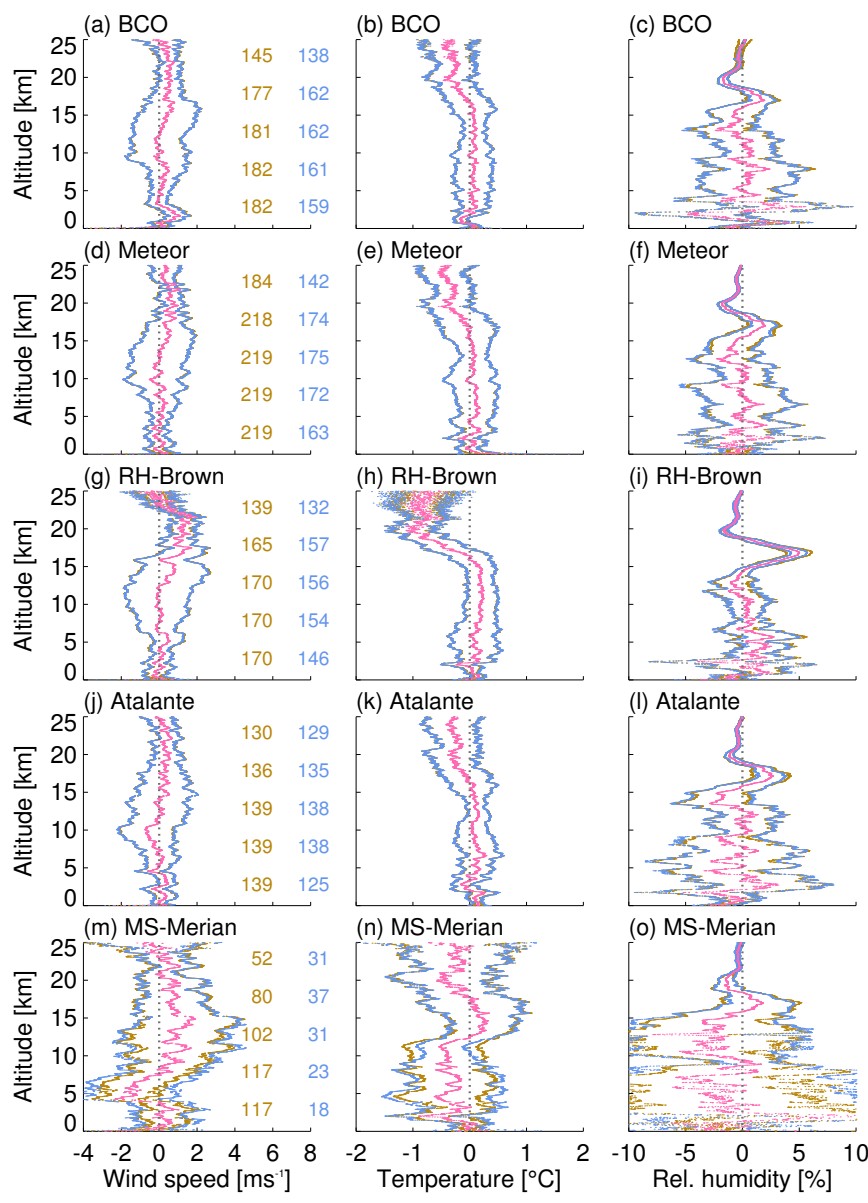

**Figure 6.** Comparison of (left) horizontal wind speed, (middle) air temperature, and (right) relative humidity, measured during ascent and descent. The pink dots show the time-averaged values during ascent minus the time-averaged values during descent. Brown (blue) dots show the 95 % confidence intervals for ascent (descent). Numbers inside the panels on the left-hand side show the counts of ascending (brown) and descending (blue) radiosondes that crossed the corresponding height-levels (2.5, 7.5, 12.5, 17.5 and 22.5 km, respectively.)

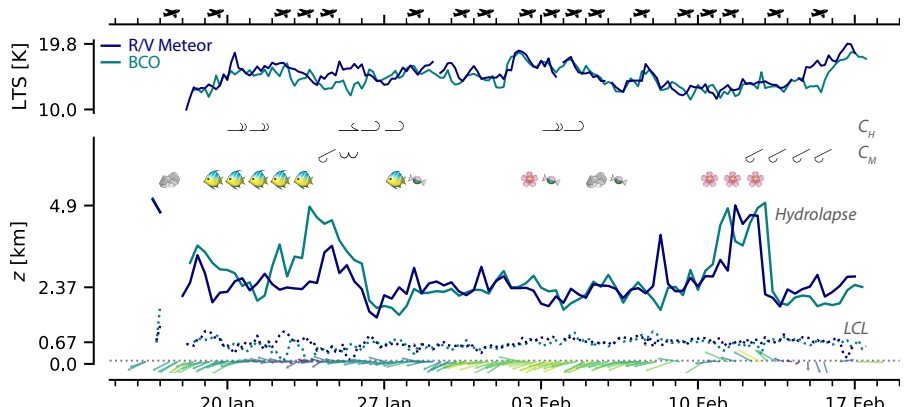

**Figure 7.** Synoptic overview of period and region of intensive aircraft measurements. Plotted is the potential temperature at $700\,\mathrm{hPa}$, the height of the hydrolapse, defined as the mean height of a running $500\,\mathrm{m}$ range in which mean relative humidity first drops below $30\,\%$, the lifting condensation level (LCL) and the wind vector averaged over the lower $200\,\mathrm{m}$. Winds are $12\,\mathrm{h}$ median values, other quantities are resampled on a $4\,\mathrm{h}$ interval, with median values plotted except for the LCL where minimum values are plotted. For the wind vectors the maximum and minimum wind speeds are $12.3\,\mathrm{m\,s}^{-1}$ and $2.0\,\mathrm{m\,s}^{-1}$, respectively. Tick marks denote maximum and minimum $\theta_{700}$, and maximum and median height of *Meteor* hydrolapse and the mean height of the LCL (*Meteor*). Also shown are days when aircraft with dropsondes were flying, the synoptic cloud observations of mid-level ($C_M$) and high ($C_H$) clouds with the associated WMO cloud-sybmol (Table 14 of 2017 World Meteorological Organization Cloud Atlas, https://cloudatlas.wmo.int/en/home.html) that predominated for that day. Cloud types are taken from the Barbados Meteorological Service SYNOP reports. Days on which a mesoscale pattern of shallow convection, following the classification activity of Schulz (2020b), was readily identified are indicated by the emojis for Fish, Sugar (candy), Flowers or Gravel (rocks).

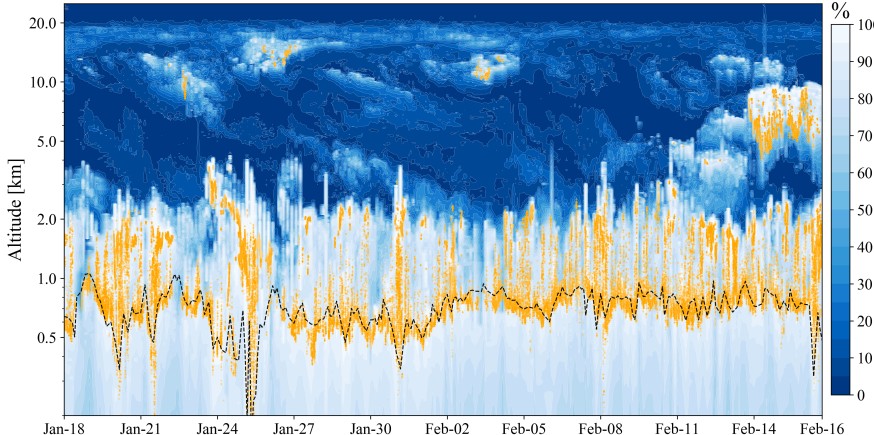

**Figure 8.** Comparison between ascending soundings and ceilometer measurements on the *Meteor*. The relative humidity from radiosonde measurements is shown in blue-to-white shading. The black dashed line represents the lifting condensation level calculated based on Bolton (1980). Cloud base heights as observed by the ceilometer are marked with orange dots. The vertical axis is chosen to be logarithmic for better visibility of the moisture distribution near the surface. The time-axis for the soundings uses launch time. The temporal resolution of the ceilometer data is 10 s. Low-altitude relative humidity profiles (300 m to 800 m) of the descending soundings were recovered by assuming a dry adiabat temperature and a constant humidity profile.

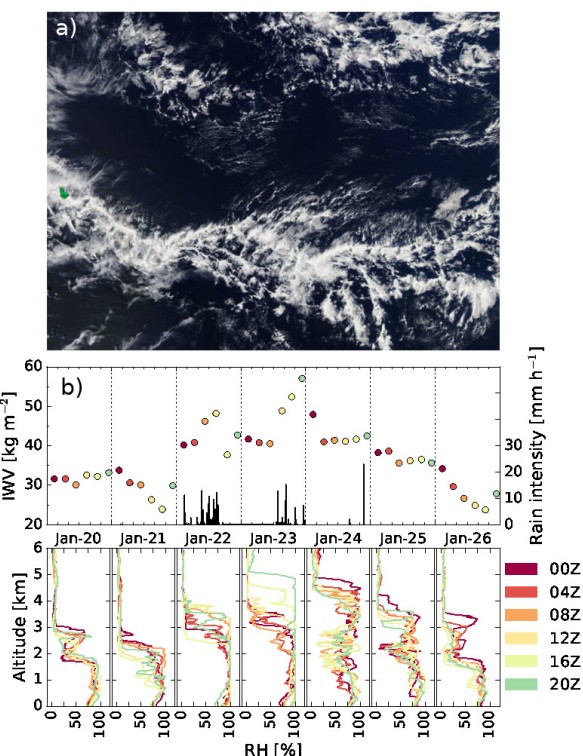

**Figure 9.** Fish cloud pattern passing Barbados between January 22–24, 2020. (a) MODIS-Aqua scene from January 22. The image covers 9–18 °N, 48–60 °W with Barbados shown in artificial green. (b) Temporal evolution of relative humidity (lower panel) and integrated water vapor (IWV; upper panel, color-coded) as measured by the BCO soundings January 20–26. Profiles and calculated IWV values are color-coded according to the nearest hour of the sounding reaching 100 hPa. The upper panel also shows a one-minute running mean of rain intensity recorded at BCO (black).

Earth System
Science
Data

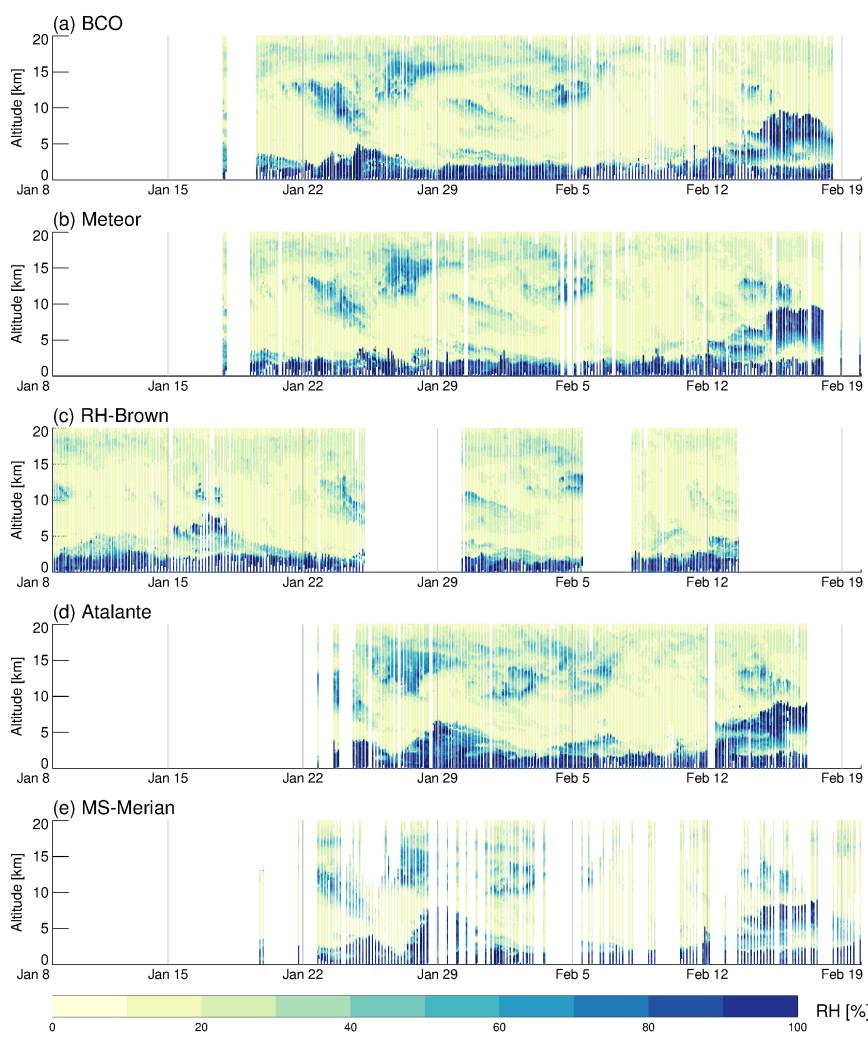

**Figure 10.** Time-height series of relative humidity measurements from all platforms. The plot combines ascending and descending soundings.

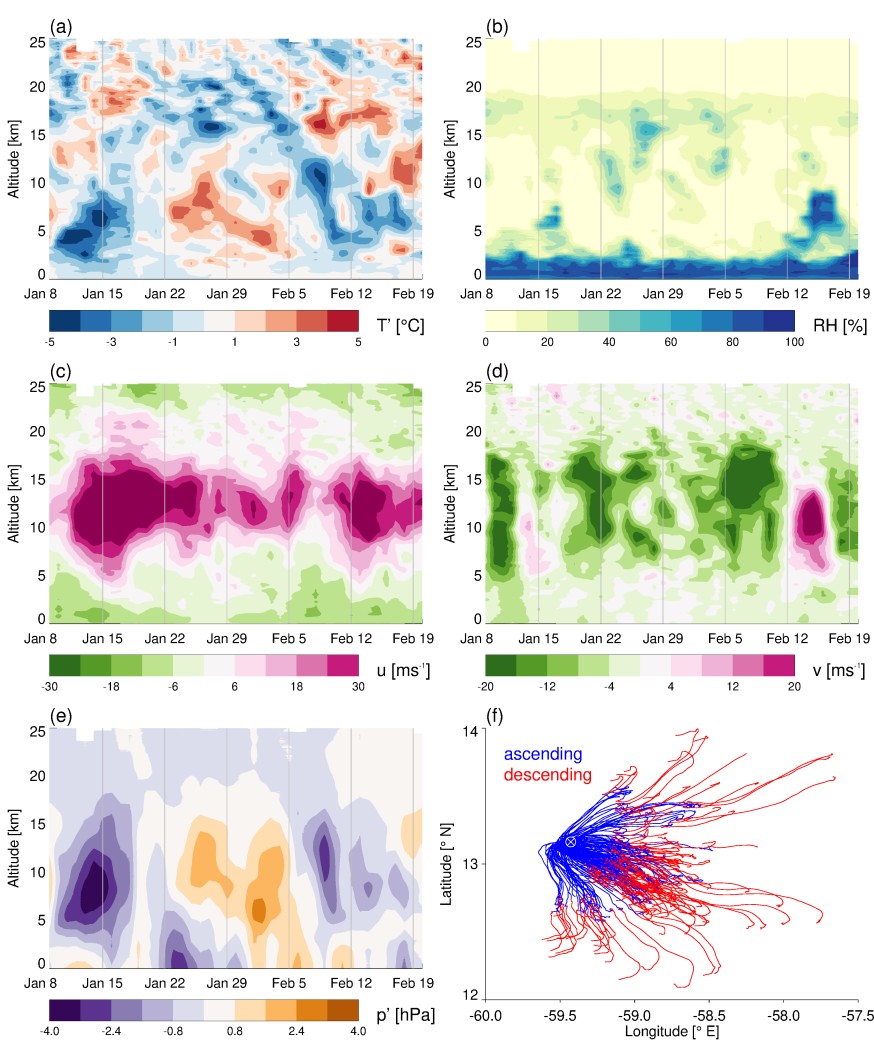

**Figure 11.** (a-e) Time-height cross sections of daily (a) temperature anomaly, (b) relative humidity, (c) zonal wind, (d) meridional wind and (e) pressure anomaly computed from ascending soundings north of 12.5 °N. The data combine 182 soundings from the BCO, 169 from the *Brown*, 150 from the *Meteor*, 28 from the *Merian* and 4 from the *Atalante*. Anomalies are defined as deviations from the time average at each altitude. (f) The horizontal trajectories of ascending and descending, respectively, radiosondes launched from the BCO.



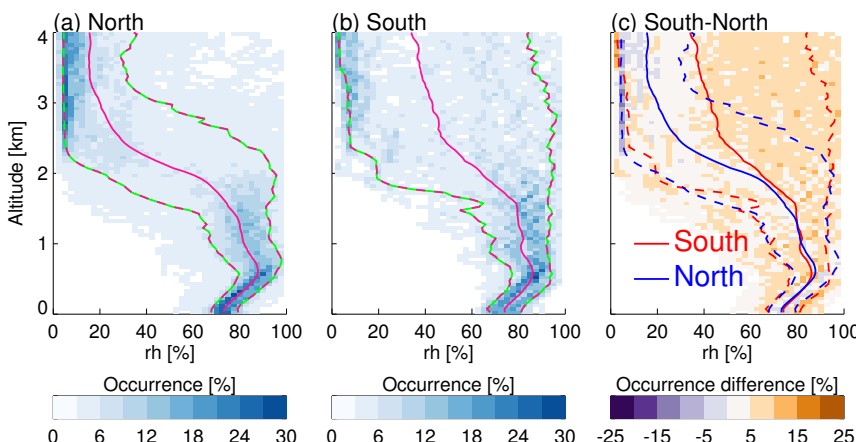

**Figure 12.** Occurrences of relative humidity as a function of height below 4 km for all soundings launched between January 26 and February 12 (437 profiles). The sum of occurrence frequencies in each altitude bin is 100 %. Altitude bins are 50 m deep and each $x$-axis contains 40 bins. North (panel a) designates soundings from the northern (12.5–14.5 °N; 261 profiles) latitude band, and South designates soundings from southern (8.5–10.5 °N, 63 profiles) latitude band. Solid lines show the mean profiles in each region and dashed lines the 10th and the 90th percentiles. Only data from ascending radiosondes are used in this comparison.



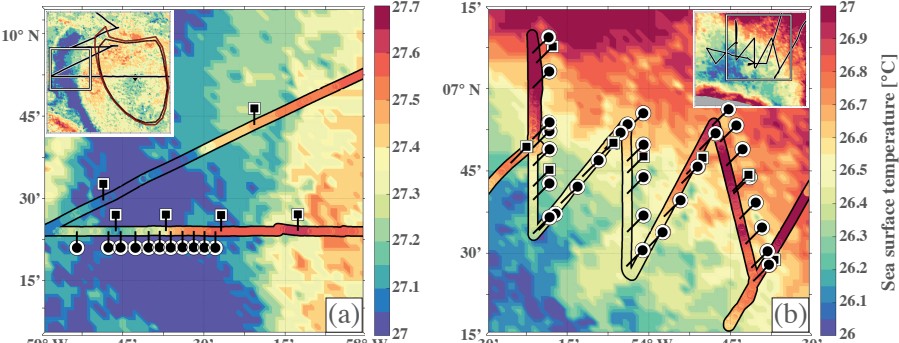

**Figure A1.** Maps of CLS SST (°C) for (a) January 25, 2020, and (b) February 2, 2020, with the *Atalante* track during the first (January 26) and second (February 2–3) intensive leg, respectively. The color shows the SST measured by the ship's ThermoSalinoGraph (TSG) at 5 m depth and the ticks show the location of Vaisala (squares) and MeteoModem (circles) radiosonde launches. Inserts in the upper corners, where the black lines indicate the ship's course, show the larger scale view of the corresponding scenes with the geographical imprint indicated by white squares. In the panel insert a, the closed contours and the black diamond indicate, respectively, the edges of an anticyclonic eddy and the position of its center.



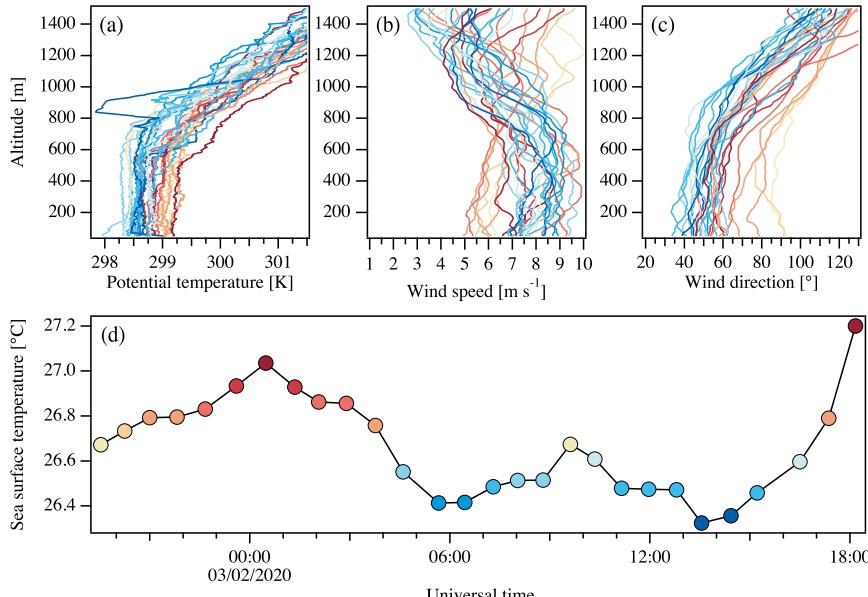

**Figure A2.** Vertical profiles (50–1500 m) from MeteoModem M10 sondes launched during the second targeted intensive radiosonde period (Figure A1b) for (a) potential temperature, horizontal wind (b) speed and (c) direction, and (d) the corresponding SST time series from the *Atalante* TSG with each circle corresponding to a MeteoModem launch. Colors are indicative of the SST (°C) at the time of each launch. Vertical profiles are built from Level-0 raw measurements.



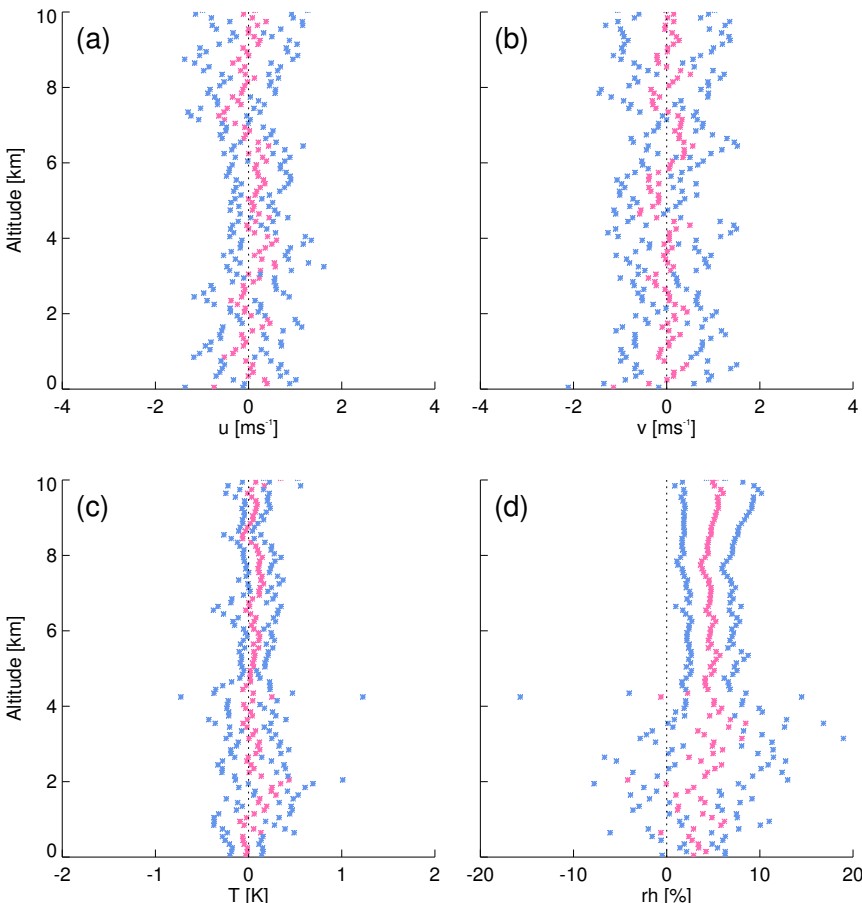

**Figure A3.** For *Atalante* soundings launched within ±25 min, the mean difference MeteoModem-Vaisala (pink) and ±1 standard deviation (blue) computed on 8 difference profiles with a vertical resolution of 100 m. Shown are difference profiles for (a) zonal wind, (b) meridional wind, (c) temperature, and (d) relative humidity.

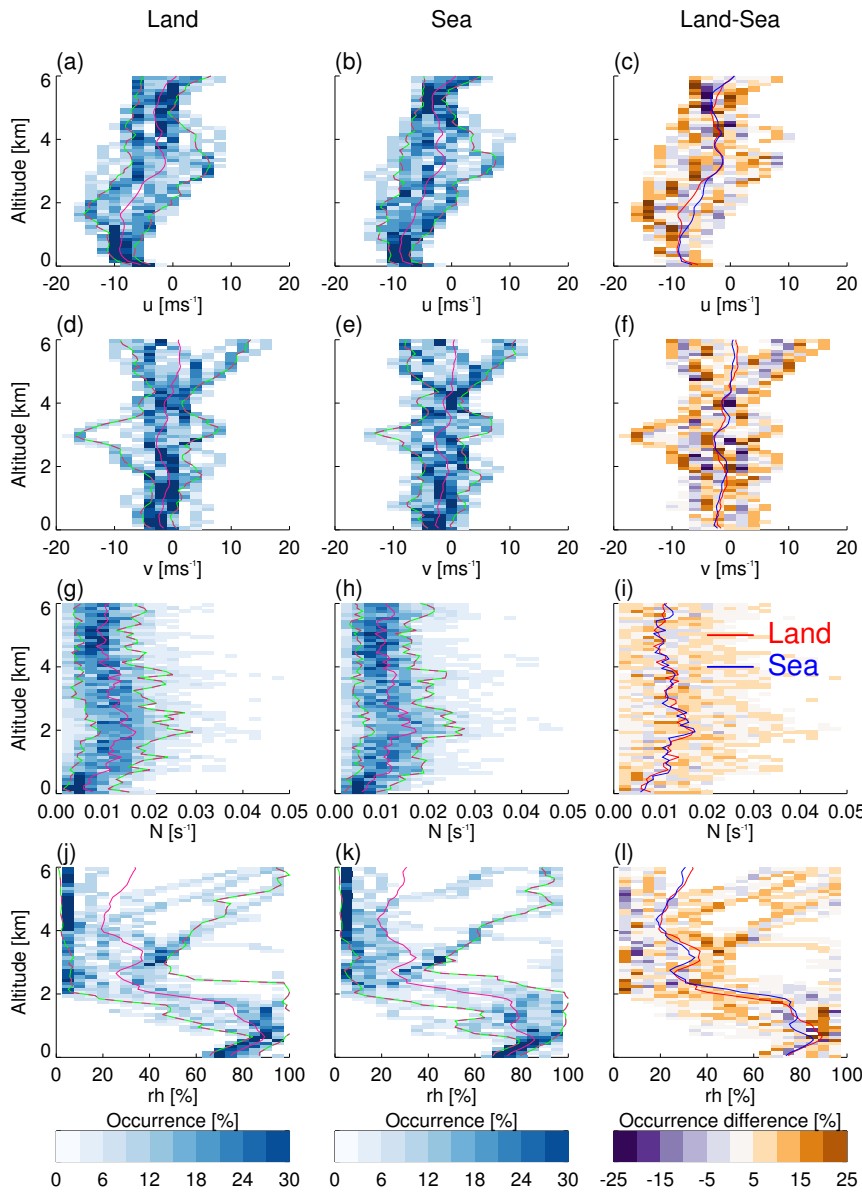

**Figure A4.** As Fig. 12, but instead of comparing different regions, we here compare ascending soundings launched from BCO with ascending soundings launched within $\pm 90\,\mathrm{min}$ from nearby ships (within $1°$ longitude to the east and $\pm 1°$ latitude of BCO, resulting in 12 matching soundings). Altitude bins are $100\,\mathrm{m}$ deep and there are 20 bins on the $x$-axis.



**Table 1.** For each platform the rows list (1) the numbers of recorded ascending soundings, (2) the numbers of recorded descending soundings, (3) the first date of data coverage, (4) the last date of data coverage, (5) whether or not parachutes were used, (6) the station altitude relative to sea level, (7) the GPS antenna offset relative to the station, (8) the launch site offset relative to the station, (9) the surface barometer offset relative to the station, (10) the frequency used to transmit the signal from the radiosonde to the antenna.

| | BCO | *Meteor* | | *Brown* | *Atalante* | *Merian* |
|---|---|---|---|---|---|---|
| | | MPI-M | DWD | | | |
| Number ascents | 182 | 180[1] | 39[1] | 170 | 139 | 118 |
| Number descents | 162 | 175[1] | - | 159 | 138 | 38 |
| Start date | Jan 16 | Jan 16 | Jan 18 | Jan 8 | Jan 21 | Jan 18 |
| End date | Feb 17 | Mar 1 | Feb 26 | Feb 12 | Feb 16 | Feb 19 |
| Use of parachutes | yes | yes | no | no | yes | yes |
| Station altitude (msl) | 25.0 | 16.9 | 5.4 | 4.3 | 13.1 | 10.4 |
| GPS antenna offset (m) | 4.3 | 2.5 | 3 | 5.5 | 2.6 | 1.6 |
| Launch site offset (m) | 0.0 | -11.5, -14.2[2] | 0.0, -2.7[2] | 0.5 | -0.6 | 0.0 |
| Surface barometer offset (m) | 1.0 | -16.9 | -5.4 | -4.3 | 0.2 | 0.6 |
| Frequency (MHz) | 400.2 | 401.5 | 403.0 | 400.5 | 400.7 – 401.2 | 402.0 |

[1] includes 8 additional soundings after Feb 20, 0 UTC
[2] Feb 9, 18 UTC - Feb 20, 0 UTC