# Peer review of "Ship- and island-based atmospheric soundings from the 2020 EUREC4A field campaign"

_Earth System Science Data, 2020_

## Referee Comment (RC1) · Anonymous Referee #1 · 9 Sep 2020

Summary.

The manuscript "Ship- and island-based atmospheric soundings from the 2020 EU-REC4A field campaign" by Stephan et al. describes the experimental design, measurements, and post-processing of the radiosonde program at EUREC4A, which was distributed across five platforms east of Barbados in January and February 2020. The manuscript is well-written and concise. It provides useful documentation for users of the data sets and is appropriately scoped for ESSD. The description of the level 2 data is disappointing as it does not appear to include additional quality control expected for this stage of processing that could have been implemented using available software,

like ASPEN. I also have some questions about the data acquisition and processing because it is not clear if the baseline corrections were implemented or omitted. The answers to these questions may only require some clarifying statements, or perhaps a more significant revision of the data set is needed.

Major Comments.

(1) You state that the descent data was collected. I am not certain if this applies to MW41, but for older Vaisala systems, collecting descent data meant running in "research mode", which does not include the standard corrections for solar heating or pendulum motions and omits some quality control procedures too. It is therefore not clear to me if this standard Vaisala processing is included in level 1 or not. If it is, please clarify. If it is not, the data likely needs to be reprocessed.

(2) Descent data is subject to some well-documented biases, some of which I noted above. While Figure 6 and discussion provides good documentation of the relative differences observed between up and down data at EUREC4A, the presentation implies the two types of data are equitable when they are not. It should be made clear that the confidence in the ascent data is higher and you should describe the limitations of the descent data.

(3) Level 2 data: ASPEN is mentioned at Line 205, but it is not defined or referenced. The roadmap provided by Ciesielski et al. (2012, https://doi.org/10.1175/BAMS-D-11-00091.1), which you state that your data is consistent with (Line 180), suggests this step is necessary for a level 2 data set, but it is not clear if this algorithm (or similar) was applied to the level 2 data or not. What were the quality control procedures applied in level 2? It is also important to consider additional sources of error: e.g., even with sonde equilibration, previous experiments on the Brown have identified biases associated with localized heating of the sonde by the ship's superstructure during equilibration, pressure errors associated with relative wind direction, spurious data caused by the ship's exhaust, and other problems (Hartten et al. 2018, www.earth-syst-sci-

data.net/10.1165/2018/). Have you considered any of these potential sources of error and if so how did you address them?

Minor Comments.

Introduction: Consider adding a statement clearly indicating that the current manuscript addresses only the surface-based radiosonde program and not the dropsonde program, which are implied to be closely linked in the experimental design of EUREC4A. Also, if you have it, cross-referencing the doi and or paper describing the dropsondes somewhere in the manuscript would be helpful to users.

Figure 2. (a) I realize the array was positioned over a featureless region of open ocean and this is essentially the map. However, more geographical context is needed for the reader. Having this map include the Caribbean Islands and the northern coast of South America would help, but perhaps some of the details of the transects would be too small after zooming out. Instead, maybe you could include a map with this figure with 6-16N and -60 - -50E displayed as a box to highlight the study region. Indeed, Fig. A1, which includes the inset as well as a useful pattern of SST as a backdrop is an improvement over Figure 2. (b) Can you mark the aircraft pattern too? (c) It would also be useful to see the location of BCO on a map of the island with the prevailing wind direction and maybe the drift tracks of the sondes launched from that station (e.g., context for Lines 247-248)

Lines 87-88: (a) This sentence is confusing as written. I think you mean that you launched 6 times per day (every 4 hours) and that this schedule included 2 launches per day that were timed to match the 0 and 12 Z synoptic times. (b) Those synoptic-schedule times would be the 10:45 and 22:45 launches, but this seems early. Normally for a 90 min launch (Line 82) you would launch 45 min early, so 11:15 and 23:15. Is there an explanation for this?

Section 2.1: (a) It is not clear if the operators from the platforms followed an agreed-upon standard set of operational procedures or if they acted independently. For example, was the balloon filling amount consistent? Was the balloon size the same? Was the equilibration procedure consistent? Was the met station and use of met data consistent? (b) There was apparently a large temperature difference between the labs where the sondes were prepared and the release point outside, yet only the Brown's procedures note an equilibration period on deck. If the other locations did not equilibrate the sondes, please note this and provide a warning about the potential for thermal instabilities or shock in the lower atmosphere within the data set.

Section 2.2: (a) Please provide the WMO station ID numbers used for the GTS in the text or table for all platforms. (b) Were all soundings sent to GTS or only the subset on the 6 or 12 hour standard schedule?

Section 2.3.2: (a) Are the bin heights centered, top, or bottom of the averages?

Line 215: change "smaller" to "slower"
* * *

---

## Referee Comment (RC2) · Anonymous Referee #2 · 17 Sep 2020

The manuscript by Stephan et al. describes the radiosonde data set obtained during the EUREC4A field campaign. Weather balloons were launched from 4 ships and one island station at Barbados. All data were obtained using the Vaisala radiosonde system. The paper describes the setup, launch operations, data collection, and processing. The three levels of data processing are publicly available.

Some analysis of the data set demonstrates its huge potential and usefulness for atmospheric science.

The raw data are complete, the level 1 and level 2 data are ncdf files following the CF convention and appear to be properly formatted.

[Figure]

The paper is overall well written and the data set overall well documented. However, there are a number of smaller issues, which should be addressed. I would recommend publication after some corrections.

Detailed comments:

The data set of the Meteor contains a set of corrected raw data files. Can you elaborate why a correction was needed and how it was applied?

Lines 80: Please describe here down to what altitudes descent data were typically recorded. This is given later in the manuscript, but should be moved to here.

Lines 81: add "... nearly match fall speeds in the middle and lower troposphere to balloon ascent speeds".

Line 82: "somewhere above each platform": Could you please make a statement about the drift of the soundings over the vertical region of interest. Something like the average horizontal distance between ascent and descent measurement at a relevant altitude.

Line 83: Which software version of the MW41 system was used at each station?

Since the RS41 SGP sondes were used, I assume the reported pressures are the measured pressures. Since the Modem sondes used GPS height for that purpose, it might be useful to highlight that difference.

Line 95: add "... which sometimes delayed soundings ...".

Throughout the description of the balloon filling at the different platforms, it was not very clear, how the amount of fill gas was gauged. Do I understand correctly, that the volume was estimated based on for example a marker (e.g. R/V Meteor) inside the container? Or was there some attempt to measure the amount of gas by monitoring the gas pressure, explicitly measuring the volume or measuring lift? Please clarify.

What balloon size/sizes were used at the different stations? On soundings with parachutes, were balloons with internal parachutes used, or were parachutes added

externally? If this was different on the Meriam, it could possibly explain their larger number of faster falling sondes.

Lines 118: The nighttime soundings during leg 1 used less helium to increase the vertical resolution. This was changed after that. Was there no value in doing so? Was anything useful learned? The authors could briefly explain this change.

Line 231: Figure 11 is referenced before Figure 10, which needs to be corrected.

Line 106f, 122ff, 133 ff, 155ff and lines 207ff: The influence of a ship on observations near the surface is well documented and understood. It is not very clear how this was handled here. I understand that the operators tried to minimize that effect by launching from a location on a ship that minimizes this effect to the extent possible. Was any additional data screening done to evaluate and filter the effect of a ship in the lowest 50 m or so? The Vaisala system has a setting that filters out data showing a superadiabatic lapse rate. If that setting was used, then some of the ships influence may be filtered out by the Vaisala system in their level 1 data.

The setup of sounding systems on ships can be tricky, since launch site, receiving antenna, reference pressure sensor, and wind measurements may be several 10s of m separated vertically. Have you verified that the altitudes in the lowest 50 m are all reasonable and consistent? I noticed that there may be unreasonable jumps of more than 20 m in the first second after launch.

I checked the altitude of the BCO launch site, which is set at 25 m in the files, but may only be 13 m in reality. It's not clear at what level above msl the balloons were launched from the ships.

Maybe the authors could comment how important the lowest 50 m of profile are in their studies. I guess they were not very important, but offsets like 10 may shift the entire profile and may be significant in high resolution studies.

Lines 230ff: I assume that the difference in atmospheric conditions downwind and an

hour later would increase the scatter between ascent and descent measurements, but should not cause any systematic bias. A different sensor response (including GPS) between ascent and descent is more likely to cause systematic biases.

Line 255: Please move the definition of the hydrolapse from the legend of Figure 7 to here.

Lines 270ff: The patterns Sugar, Gravel, Flower and Fish are not obvious and scientifically accepted patterns. Please describe these here.

Line 351: Change to "... together with the NRT SST maps produced by ...". Can you add, which satellite(s) is used for these maps?

Line 384: Was there a particular reason to use the Magnus-Tetens formula for the M10 sonde humidity calculations? This could have been handled the same as the Vaisala sondes. However, I do not expect that the differences are significant over the region of interest.

Figure 5: The spread of the rise rate appears a little large, in particular in the stratosphere, where the balloon rise rate becomes a lot more uniform. I don't believe that ascent rates are calculated on 500 m bins, rather I assume that the 1 s calculated rise rates were binned in 500 m bins. The spread shown in this Figure is most likely due to noise in the pressure data. If the rise rate was calculated based on GPS altitude or better still based on 500 m altitude bins, the spread should decrease significantly. I do not think it is necessary to redo this plot, but it would be good to explain the spread.

Figure 6: What does "time averaged" mean in this Figure? If the same time averaging was used on ascent and descent and then the data were binned to consistent altitudes, I would not be surprised of biases due to the different descent rate profile compared to ascent. However, that may not be the case. Please clarify.

Figure 7: What is the axis label "LTS [K]"? I assume this is the 700 hPa potential temperature, but the axis label indicates something else.

[Figure]

Figure 8: The legend refers to humidity profiles on descent below the last received data. However, the Figure does not refer to descent measurements. This sentence can probably be deleted.

Figure 11 f shows some trajectories but does not fit with the rest of the panels and is not described in the text. This panel could be made a standalone figure and address my point regarding the average drift.

Figure 12: The legend indicates that this Figure shows 437 profiles, but the sum of North and South does not add up to that number. More than 100 profiles seem to be missing.

The appendix describes some results of the Modem radiosonde launches. I would suggest to add a few sentences to the general data processing similar as the description of the Vaisala data. Was all data QC done by the Modem software? Were similar data levels (0/1/2) generated?

---

## Author Comment (AC1) · 23 Dec 2020

Response to Reviewers' comments

We would like to thank the two anonymous reviewers for their detailed assessment of our manuscript and the associated data. Their comments were very valuable for improving both. We addressed all remarks, as detailed below. Line references refer to the tracked-changes document. The tracked-changes document and a pdf file of this response with our replies marked in color can be found in the attached zip folder.

Reviewer #1

Summary. The manuscript "Ship- and island-based atmospheric soundings from the 2020 EUREC4A field campaign" by Stephan et al. describes the experimental design, measurements, and post-processing of the radiosonde program at EUREC4A, which was distributed across five platforms east of Barbados in January and February 2020. The manuscript is well-written and concise. It provides useful documentation for users of the data sets and is appropriately scoped for ESSD. The description of the level 2 data is disappointing as it does not appear to include additional quality control expected for this stage of processing that could have been implemented using available software like ASPEN. I also have some questions about the data acquisition and processing because it is not clear if the baseline corrections were implemented or omitted. The answers to these questions may only require some clarifying statements, or perhaps a more significant revision of the data set is needed.

We thank the reviewer for their positive feedback. We clarified the information on data acquisition, treatment and quality control applied to Level 2. Details are listed below.

Major Comments

(1) You state that the descent data was collected. I am not certain if this applies to MW41, but for older Vaisala systems, collecting descent data meant running in "research mode", which does not include the standard corrections for solar heating or pendulum motions and omits some quality control procedures too. It is therefore not clear to me if this standard Vaisala processing is included in level 1 or not. If it is, please clarify. If it is not, the data likely needs to be reprocessed.

The MW41 software processes the descending phase of a sounding in the exact same way as the ascending phase. No "research mode" was turned on. We clarified this important point at line 224: "The MW41 software applies the same correction and quality control steps to the descending and ascending phases of a sounding."

(2) Descent data is subject to some well-documented biases, some of which I noted above. While Figure 6 and discussion provides good documentation of the relative differences observed between up and down data at EUREC4A, the presentation implies the two types of data are equitable when they are not. It should be made clear that the confidence in the ascent data is higher and you should describe the limitations of the descent data.

We clarified that descent data are associated with greater uncertainties by adding to the quality control section (2.3.1) at line 225: "Descending sondes, however, can be subject to uncontrollable factors. For example, a falling device may be affected by the remaining debris of a balloon. For this reason, Vaisala does not guarantee the same above-mentioned error margins for data from descending soundings."

The discussion of Fig. 7 (old Fig. 6) is now motivated in the following way (line 263): "Despite corrections and quality control steps applied by MW41, measurements taken during descent may be accompanied by larger uncertainties due to less favorable and more variable measurement conditions. To establish what degree of confidence we may attribute to the descent data, Fig. 7 compares the measurements of horizontal wind speed, air temperature and relative humidity between ascending and descending soundings."

(3) Level 2 data: ASPEN is mentioned at Line 205, but it is not defined or referenced. The roadmap provided by Ciesielski et al. (2012, https://doi.org/10.1175/BAMS-D-11-00091.1), which you state that your data is consistent with (Line 180), suggests this step is necessary for a level 2 data set, but it is not clear if this algorithm (or similar) was applied to the level 2 data or not. What were the quality control procedures applied in level 2?

The treatment of the data by the MW41 software is equivalent to the quality control procedures that ASPEN applies. Reference to ASPEN and clarification of this aspect are now given at line 235: "..., which is also used by the Atmospheric Sounding Processing ENvironment (ASPEN) software (Suhr and Martin, 2020) for EUREC4A dropsonde measurements. Surface-launched soundings were not reprocessed with ASPEN, as

the ASPEN manual warns against duplicating quality control procedures applied by the Vaisala MW41"

It is also important to consider additional sources of error: e.g., even with sonde equilibration, previous experiments on the Brown have identified biases associated with localized heating of the sonde by the ship's superstructure during equilibration, pressure errors associated with relative wind direction, spurious data caused by the ship's exhaust, and other problems (Hartten et al. 2018, www.earth-syst-sci-data.net/10.1165/2018/). Have you considered any of these potential sources of error and if so how did you address them?

This is the reason why we set values below 40 m to missing in the Level-2 data. We now state this explicitly at line 239: "Discarding the lowest 40 m avoids potential biases in the soundings associated with local ship effects, like heating or exhaust plumes, and other problems that are discussed by, e.g. Hartten et al. (2018)."

Minor Comments

Introduction: Consider adding a statement clearly indicating that the current manuscript addresses only the surface-based radiosonde program and not the dropsonde program, which are implied to be closely linked in the experimental design of EUREC4A. Also, if you have it, cross-referencing the doi and or paper describing the dropsondes somewhere in the manuscript would be helpful to users.

We agree that the introduction may have given the wrong idea about the focus of this paper. We moved and modified the following sentence (line 40): "This article introduces the radiosonde observations and their resulting data sets. Other measurements, including the dropsonde data, are described in the overview paper by Stevens et al. (2020) and the references therein."

The dropsonde data paper has not been submitted at the time of this resubmission. We hope to add the reference to the final version of this manuscript.

Figure 2. (a) I realize the array was positioned over a featureless region of open ocean and this is essentially the map. However, more geographical context is needed for the reader. Having this map include the Caribbean Islands and the northern coast of South America would help, but perhaps some of the details of the transects would be too small after zooming out. Instead, maybe you could include a map with this figure with 6-16Nand -60 - -50E displayed as a box to highlight the study region. Indeed, Fig. A1, which includes the inset as well as a useful pattern of SST as a backdrop is an improvement over Figure 2.

We added the island of Barbados and the South American coastline to Fig. 2. We agree that this modification is helpful and thank the reviewer for suggesting it.

(b) Can you mark the aircraft pattern too? We appreciate this idea to give context to the ship tracks. The aircraft pattern is now marked in Fig. 2c and we refer to it at line 75.

(c) It would also be useful to see the location of BCO on a map of the island with the prevailing wind direction and maybe the drift tracks of the sondes launched from that station (e.g., context for Lines 247-248)

We thank the reviewer for this remark. In following the suggestion by reviewer #2 (marked by ˆˆ below in this document), we made the panel showing the sonde tracks a stand-alone figure (now Fig. 3). The figure also shows the location of the BCO on Barbados and the prevailing wind direction at 500 m altitude derived from the soundings. We refer to the new figure at lines 86, 109 and 268.

Lines 87-88: (a) This sentence is confusing as written. I think you mean that you launched 6 times per day (every 4 hours) and that this schedule included 2 launches per day that were timed to match the 0 and 12 Z synoptic times.

We clarified this sentence by rephrasing to (line 96): "The default launch times were 0245, 0645, 1045, 1445, 1845, and 2245 UTC. This schedule was selected to include

two launches per day that were timed to match the 00 and 12 UTC synoptic times."

(b) Those synoptic-schedule times would be the 10:45 and 22:45 launches, but this seems early. Normally for a 90 min launch (Line 82) you would launch 45 min early, so 11:15 and 23:15. Is there an explanation for this? We clarified the timing of the launches by adding at line 98: "In practice the soundings reached 100 hPa on average in 60 minutes and burst after 90 minutes." The precise timing of soundings used to be more critical for numerical weather prediction. Recent advances, including 4D-Var assimilation, allow for more flexibility. Moreover, preferred reporting times differ between GTS entry points.

Section 2.1: (a) It is not clear if the operators from the platforms followed an agreed-upon standard set of operational procedures or if they acted independently. For example, was the balloon filling amount consistent? Was the balloon size the same? Was the equilibration procedure consistent? Was the met station and use of met data consistent?

The balloon sizes are now listed: "Vaisala sondes were attached to 200 g balloons (BCO, Atalante, Merian, Meteor) or 150 g balloons (Brown). When present, the balloons were equipped with internal parachutes (see Table 1 for the use of parachutes). A modification took place on the Atalante, where after 0800 UTC on February 8, 350 g balloons with external parachutes were used instead." (line 89) "In addition to the Vaisala soundings, 47 sondes of MeteoModem type M10 attached to 150 g balloons without parachutes were launched. . ." (line 153).

We added a summary before describing the details for each platform (line 101): "In the following section, we describe specific issues and aspects of the launch procedure and surface equipment particular to each platform. All stations followed best practices for different equipment, which were established by several experienced teams at in-person sounding orientations prior to the campaign. For instance, every platform used a different empirical way of gauging the fill amount of gas, to arrive at desired ascent

rates. Equipment and procedures differed between the platforms, but this does not introduce systematic biases to Level-2 data, as these data only start at 40 m height (see Section 2.3.2), where measurements are independent of the surface procedures.."

(b) There was apparently a large temperature difference between the labs where the sondes were prepared and the release point outside, yet only the Brown's procedures note an equilibration period on deck. If the other locations did not equilibrate the sondes, please note this and provide a warning about the potential for thermal instabilities or shock in the lower atmosphere within the data set.

Such shocks would be filtered out by the software, already at Level-1. We now provide this information in Section 2.3.1 (line 223): "Periods of super-adiabatic cooling are interpolated, and this also applies to temperature differences right above the surface." We mention in the description of procedures (Sections 2.1.1 to 2.1.5) which platforms released sondes out of a container. For all other platforms the open-air release provides ample time for equilibration given that the response time of the instruments is only a few seconds (see paragraph at line 215). Attaching the sonde to a balloon already takes longer than several multiples of this time. With regard to the container releases, local abrupt changes in measured profiles could exist below 40 m. These near-surface problems are discussed at line 239, and to avoid them we set Level-2 data below 40 m to missing values.

Section 2.2: (a) Please provide the WMO station ID numbers used for the GTS in the text or table for all platforms.

We added this information to Table 1 and refer to it in the text at line 190: "The WMO station identifiers and designators for tracking the data within the GTS are listed in Table 1 for each station."

(b) Were all soundings sent to GTS or only the subset on the 6 or 12 hour standard schedule?

We clarified the sentence at line 183: "...we aimed to disseminate as much of the full 1-second resolution radiosonde data from the EUREC4A campaign as possible over the GTS, regardless of the launch time."

Section 2.3.2: (a) Are the bin heights centered, top, or bottom of the averages?

The bins are centered, as we now state at line 233. We now also give their bounds in the level 2 data set (variable alt_bnds).

Line 215: change "smaller" to "slower"

We changed this (line 250).

Reviewer #2

The manuscript by Stephan et al. describes the radiosonde data set obtained during the EUREC4A field campaign. Weather balloons were launched from 4 ships and one island station at Barbados. All data were obtained using the Vaisala radiosonde system. The paper describes the setup, launch operations, data collection, and processing. The three levels of data processing are publicly available. Some analysis of the data set demonstrates its huge potential and usefulness for atmospheric science. The raw data are complete, the level 1 and level 2 data are ncdf files following the CF convention and appear to be properly formatted. The paper is overall well written and the data set overall well documented. However, there are a number of smaller issues, which should be addressed. I would recommend publication after some corrections.

We are grateful to the reviewer for their careful check of our published data. The issues they raised helped to improve the data set as well as the manuscript.

Detailed comments: The data set of the Meteor contains a set of corrected raw data files. Can you elaborate why a correction was needed and how it was applied?

** We thank the reviewer for pointing this out and added an explanation to section 2.1.2 at line 126: "By mistake, the heights of the pressure sensor, the GPS antenna

and the launching altitude were incorrectly entered at the beginning of the cruise. In addition, we noticed large delays between the time at which surface measurements were entered and the launch. Therefore, we reprocessed the raw data using the MW41 software, after correcting the sensor heights and surface data in the raw files. This post-processing is lossless and the reprocessed data have the same quality standard as the data from the other platforms. We included both the original and reprocessed Level-0 data in the dataset."

Lines 80: Please describe here down to what altitudes descent data were typically recorded. This is given later in the manuscript, but should be moved to here.

We moved the text to line 81.

Lines 81: add "...nearly match fall speeds in the middle and lower troposphere to balloon ascent speeds".

This is a very sensible correction. We modified the sentence at line 84 accordingly.

Line 82: "somewhere above each platform": Could you please make a statement about the drift of the soundings over the vertical region of interest. Something like the average horizontal distance between ascent and descent measurement at a relevant altitude.

This is nicely visualized in Fig. 3. We rephrased the text at line 84 to: "Given that a typical ascent takes about 90 min, a radiosonde was sampling the air somewhere close to each platform nearly continuously during regular operation. The horizontal drift of the sondes is shown in Fig. 3 for the example of the BCO."

Line 83: Which software version of the MW41 system was used at each station? Since the RS41 SGP sondes were used, I assume the reported pressures are the measured pressures. Since the Modem sondes used GPS height for that purpose, it might be useful to highlight that difference.

The software versions of the MW41 system were added to Table 1. These and the variables measured by the RS41 SGP sondes are introduced at line 86: "All platforms

deployed Vaisala RS41-SGP radiosondes, which measure wind, temperature, relative humidity, and pressure, and used Vaisala MW41 ground station software to record and process the sounding data. The software versions of the MW41 system are given in Table 1 for each platform. Basic algorithms and data processing did not change between these versions."

The difference to Modem sondes is made explicit at line 427: "Unlike with RS41 SGP sondes, the pressure is deduced from the altitude and the surface station pressure measurement, using the hydrostatic approximation."

Line 95: add "...which sometimes delayed soundings..."

We rephrased the sentence accordingly (line 112).

Throughout the description of the balloon filling at the different platforms, it was not very clear, how the amount of fill gas was gauged. Do I understand correctly, that the volume was estimated based on for example a marker (e.g. R/V Meteor) inside the container? Or was there some attempt to measure the amount of gas by monitoring the gas pressure, explicitly measuring the volume or measuring lift? Please clarify. What balloon size/sizes were used at the different stations? On soundings with parachutes, were balloons with internal parachutes used, or were parachutes externally? If this was different on the Meriam, it could possibly explain their larger number of faster falling sondes.

The balloon sizes are now listed: "Vaisala sondes were attached to 200 g balloons (BCO, Atalante, Merian, Meteor) or 150 g balloons (Brown). When present, the balloons were equipped with internal parachutes (see Table 1 for the use of parachutes). A modification took place on the Atalante, where after 0800 UTC on February 8, 350 g balloons with external parachutes were used instead." (line 89) "In addition to the Vaisala soundings, 47 sondes of MeteoModem type M10 attached to 150 g balloons without parachutes were launched. . ." (line 153).

We added a summary before describing the details for each platform (line 101): "In the following section, we describe specific issues and aspects of the launch procedure and surface equipment particular to each platform. All stations followed best practices for different equipment, which were established by several experienced teams at in-person sounding orientations prior to the campaign. For instance, every platform used a different empirical way of gauging the fill amount of gas, to arrive at desired ascent rates. Equipment and procedures differed between the platforms, but this does not introduce systematic biases to Level-2 data, as these data only start at 40 m height (see Section 2.3.2), where measurements are independent of the surface procedures.."

Lines 118: The nighttime soundings during leg 1 used less helium to increase the vertical resolution. This was changed after that. Was there no value in doing so? Was anything useful learned? The authors could briefly explain this change.

We added an explanation at line 140: "To avoid the potential for biasing analyses of the diurnal cycle with systematic diurnal differences in ascent rates, after January 24, the same target ascent rate was used for day and night."

Line 231: Figure 11 is referenced before Figure 10, which needs to be corrected.

We thank the reviewer for noticing this and swapped the figures.

Line 106f, 122ff, 133 ff, 155ff and lines 207ff: The influence of a ship on observations near the surface is well documented and understood. It is not very clear how this was handled here. I understand that the operators tried to minimize that effect by launching from a location on a ship that minimizes this effect to the extent possible. Was any additional data screening done to evaluate and filter the effect of a ship in the lowest 50 m or so? The Vaisala system has a setting that filters out data showing a superadiabatic lapse rate. If that setting was used, then some of the ships influence may be filtered out by the Vaisala system in their level 1 data.

This is correct. Such super-adiabatic lapse rates would be filtered out by the software,

already at Level-1. We now provide this information in Section 2.3.1, the section that describes Level-1 (line 223): "Periods of super-adiabatic cooling are interpolated, and this also applies to temperature differences right above the surface."

The setup of sounding systems on ships can be tricky, since launch site, receiving antenna, reference pressure sensor, and wind measurements may be several 10s of m separated vertically. Have you verified that the altitudes in the lowest 50 m are all reasonable and consistent? I noticed that there may be unreasonable jumps of more than 20 m in the first second after launch.

Yes, we now verified that the altitudes in the lowest part of the soundings are reasonable. Please see our response above marked by **. In addition, we added to Section 2.3.1 (line 204): "Sometimes the launch detection did not work properly, which resulted in differences of more than 30 m between the surface altitude and the first reported sonde altitude. Such profiles were reprocessed by correcting the launch time in the raw files. The files were then processed like the corrected files from the Meteor (see Section 2.1.2). "

I checked the altitude of the BCO launch site, which is set at 25 m in the files, but may only be 13 m in reality.

The height of the launch site is measured with two independent GPS devices. We can confirm that 25 m are correct for the height of the launch site.

It's not clear at what level above msl the balloons were launched from the ships. Maybe the authors could comment how important the lowest 50 m of profile are in their studies. I guess they were not very important, but offsets like 10 may shift the entire profile and may be significant in high resolution studies.

The launch heights are given in Table 1 for each platform and the lowest 40 m of ship soundings in the Level-2 data are set to missing values as described in section 2.3.2 in order to exclude influences of the ships' superstructure on the measurements.

We clarified in the caption of Table 1 that the listed altitudes and offsets refer to the waterline of the ship which is indeed not necessarily identical to mean sea level at all times.

Lines 230ff: I assume that the difference in atmospheric conditions downwind and an later would increase the scatter between ascent and descent measurements, but should not cause any systematic bias. A different sensor response (including GPS) between ascent and descent is more likely to cause systematic biases.

We thank the reviewer for pointing this out and agree with them. The text was adapted accordingly: "Despite corrections and quality control steps applied by MW41, measurements taken during descent may be accompanied by larger uncertainties due to less favorable and more variable measurement conditions. To establish what degree of confidence we may attribute to the descent data, Fig. 7 compares the measurements..." (line 263). Then at line 268: "Meridional horizontal drift could create systematic biases.", and at line 269: "Second, there are variable time lags of the order of a couple of hours between ascending and descending measurements, which we expect might increase the scatter between ascent and descent measurements but not create systematic differences. A systematically different response of the sensors during descent might be the most important factor for biases."

Line 255: Please move the definition of the hydrolapse from the legend of Figure 7 to here.

We followed this suggestion and moved the definition of the hydrolapse from the caption of Figure (now) 8 to now line 296.

Lines 270ff: The patterns Sugar, Gravel, Flower and Fish are not obvious and scientifically accepted patterns. Please describe these here.

We added the following sentence to the manuscript (line 312): "While the low and small Sugar clouds appear with little organization, Gravel clouds reach deeper extents and

organize along gust fronts. The fish-â■bone like organization of clouds on horizontal scales of 200-â■‐2000 km is described by the Fish pattern, and large stratiform, often circular-shaped cloud clumps are labeled as Flowers."

Line 351: Change to "...together with the NRT SST maps produced by...". Can you add, which satellite(s) is used for these maps?

We made the change, and added the list of satellites at line 388: "The CLS SST NRT product is derived from nighttime observations (to avoid diurnal warming of the sea surface) by the MODerate-resolution Imaging Spectroradiometer (MODIS) on board TERRA and AQUA satellites, the Advanced Very High Resolution Radiometer (AVHRR) on board METOP-A and -B, the Visible Infrared Imager Radiometer Suite (VIIRS) on board Suomi-NPP, the Advanced Himawari Imager (AHI) on board HIMAWARI-8, and the Advanced Baseline Imager (ABI) on board GOES-16 and -17."

Line 384: Was there a particular reason to use the Magnus-Tetens formula for the M10sonde humidity calculations? This could have been handled the same as the Vaisala sondes. However, I do not expect that the differences are significant over the region of interest.

We modified the processing of the MeteoModem soundings to be the same as that of the Vaisala soundings by beginning with the raw data instead of starting from the BUFR files. The text was adapted at line 430: "The raw MeteoModem data are processed in the same way as the Vaisala soundings...."

Figure 5: The spread of the rise rate appears a little large, in particular in the stratosphere, where the balloon rise rate becomes a lot more uniform. I don't believe that ascent rates are calculated on 500 m bins, rather I assume that the 1 s calculated rise rates were binned in 500 m bins. The spread shown in this Figure is most likely due to noise in the pressure data. If the rise rate was calculated based on GPS altitude or better still based on 500 m altitude bins, the spread should decrease significantly. I do not think it is necessary to redo this plot, but it would be good to explain the spread.

The reviewer is correct with their assumption about the computation. We now explain how the figure was computed at line 247: "The figure is based on the ascent (or descent) rates with a 10-m vertical resolution included in the Level-2 data." It is our intention to present the data that we publish and therefore we agree that the figure should not be changed. Regarding the spread, we decided not to discuss this further, as there are several possibilities for changes in the spread, including noise (as suggested by the reviewer), interpolation artifacts, true differences, sample size (please note that the stratosphere has a much smaller sample size!), stratospheric gravity waves. Investigating these aspects is beyond the scope of this paper. However, the interested reader has got the opportunity to investigate this question, as all required (raw) data are available.

Figure 6: What does "time averaged" mean in this Figure? If the same time averaging was used on ascent and descent and then the data were binned to consistent altitudes, I would not be surprised of biases due to the different descent rate profile compared to ascent. However, that may not be the case. Please clarify.

The wording "time averaged" was a poor choice as it is confusing. Shown is simply the average of the soundings. We rephrased the caption of (now) Fig. 7 to clarify this point.

Figure 7: What is the axis label "LTS [K]"? I assume this is the 700 hPa potential temperature, but the axis label indicates something else.

We thank the reviewer for catching this mistake. The panel shows Lower Tropospheric Stability, that is the difference of potential temperature at 700 hPa and the mean potential temperature in the lowest 200 m. The figure caption of now Figure 8 was corrected accordingly and we added the definition of the LTS to the text at line 297.

Figure 8: The legend refers to humidity profiles on descent below the last received data. However, the Figure does not refer to descent measurements. This sentence can probably be deleted. We thank the reviewer for catching this inconsistency. The

figure does in fact combine ascending and descending soundings. We corrected the first sentence in the caption (now Fig. 9) to read: "Comparison between ascending and descending soundings and ceilometer measurements on the Meteor."

ˆˆFigure 11 f shows some trajectories but does not fit with the rest of the panels and is not described in the text. This panel could be made a standalone figure and address my point regarding the average drift. We agree with the reviewer and made old Fig. 11f into the standalone figure (Fig. 3), which we now refer to at lines 86, 109 and 268. Panel 11f was replaced with a contour plot of the Brunt-Vaisala frequency.

Figure 12: The legend indicates that this Figure shows 437 profiles, but the sum of North and South does not add up to that number. More than 100 profiles seem to be missing. We thank the reviewer for catching this error and corrected the caption of (now) Fig. 13.

The appendix describes some results of the Modem radiosonde launches. I would suggest to add a few sentences to the general data processing similar as the description of the Vaisala data. Was all data QC done by the Modem software? Were similar data levels (0/1/2) generated? As mentioned above, we now process the Modem soundings like the Vaisala soundings. We give the following description at line 430: "The raw MeteoModem data are processed in the same way as the Vaisala soundings to create Level-1 and Level-2 files that match the format of the corresponding Vaisala data. The only difference is that the description of the MeteoModem corrections that are automatically applied by the software is a trade secret and therefore not known to us. However, the M10 sondes are currently in the process of being certified by the Global Climate Observing System Reference Upper-Air Network (GRUAN). If the GRUAN certification is granted, details on these corrections will become available. We checked for and corrected spurious data in the surface observations using handwritten log-sheets filed during the campaign."

Please also note the supplement to this comment:

https://essd.copernicus.org/preprints/essd-2020-174/essd-2020-174-AC1-supplement.zip